



# Persistent deep-water formation in the Nordic Seas during Marine Isotope Stages 5 and 4 notwithstanding changes in Atlantic overturning

Tim B. Stobbe[1*], Henning A. Bauch[2,3], Daniel A. Frick[1], Jimin Yu[4,5], Julia Gottschalk[1]

[1]Institute of Geosciences, Kiel University, Kiel, Germany; [2]GEOMAR Helmholtz Centre for Ocean Research, Kiel, Germany; [3]Alfred Wegener Institute Helmholtz Centre for Polar and Marine Research, Bremerhaven, Germany; [4]Laoshan Laboratory, Qingdao, China; [5]Research School of Earth Sciences, Australian National University, Canberra, Australia

*Correspondence to*: Tim B. Stobbe (tim.stobbe@ifg.uni-kiel.de)

**Short Summary:** New bottom water [$CO_3^{2-}$] reconstructions show higher levels in the deep Norwegian Sea during MIS 5 and 4 than during the Holocene. This suggests modern-like/persistent deep-water formation in this region, even when Atlantic overturning weakened and/or shoaled. Our data puts new constraints on the endmember [$CO_3^{2-}$] composition of northern component-waters emerging from the Nordic Seas, with implications for the chemical characteristics and carbon storage capacity of the Atlantic Ocean.

**Keyword group 1:** Nordic Seas, Carbonate ion, Atlantic Ocean, AMOC, last glacial cycle, epibenthic foraminiferal B/Ca

**Key word group 2:** Carbon cycle, Glacial $CO_2$ storage, deep-water formation, late Pleistocene, benthic foraminifera

**Abstract.** Reductions in the extent and formation of North Atlantic Deep Water (NADW) and the expansion of southern-sourced waters in the Atlantic Ocean were linked to enhanced marine carbon storage during glacial and stadial periods and are considered a key mechanism explaining late Pleistocene atmospheric $CO_2$ variations on glacial-interglacial and millennial timescales. However, changes in the formation of deep waters in the Nordic Seas, an important source of NADW, and their influence on the geometry and intensity of Atlantic overturning remain poorly understood, especially beyond the last glacial maximum, leaving possible impacts on atmospheric $CO_2$ changes elusive. Here, we present high-resolution *Cibicidoides wuellerstorfi* B/Ca-based bottom water [$CO_3^{2-}$] reconstructions, alongside with complementary *C. wuellerstorfi* stable oxygen and carbon isotopes and abundance estimates of aragonitic pteropods in marine sediment core PS1243 from the deep Norwegian Sea to investigate past deep-water dynamics in the Nordic Seas and potential impacts on Atlantic overturning and carbon cycling. Our data suggest continuous formation of dense and well-ventilated (high-[$CO_3^{2-}$]) deep waters throughout Marine Isotope Stages (MIS) 5 and 4, alongside a deepening of the aragonite compensation depth by at least 700 m during the MIS 5b-to-4 transition, consistent with sustained Nordic Seas convection. In addition, higher-than-Holocene bottom water [$CO_3^{2-}$] during MIS 5e highlight the resilience of Nordic Seas overturning towards a warmer North Atlantic, decreased Arctic sea ice extent and meltwater supply from surrounding ice sheets. A compilation of bottom water [$CO_3^{2-}$] records from the Atlantic Ocean indicates that dense waters from the Nordic Seas may have continuously expanded into the intermediate and/or deep (western) North Atlantic via supply of dense water overflows across the Greenland-Scotland Ridge, diminishing the capacity of the North Atlantic to store carbon during MIS 4 and stadial conditions of MIS 5. Our study emphasises differences





in the sensitivity of North Atlantic and Nordic Seas overturning dynamics to climate boundary conditions of the last glacial

cycle that have implications for the carbon storage capacity of the Atlantic Ocean and its role in atmospheric $CO_2$ variations.

## 1 Introduction

The Atlantic Meridional Overturning circulation (AMOC) is crucial for the moisture- and heat supply to the high northern latitudes, thus affecting both global climate and regional weather patterns (e.g., Rahmstorf, 2002; Adkins, 2013). Heat and moisture released by the northward flowing North Atlantic Current and the inflow of Atlantic waters into the subpolar Nordic

Seas referred to as Nordic heat pump may affect northern-hemisphere ice sheet growth (e.g., Fettweis et al., 2017; Hermann et al., 2020) and leads to mild winters in western Europe at present-day (e.g., Johns et al., 2011). Inflowing saline surface waters of Atlantic origin cool and lose buoyancy within the Nordic Seas, and along with contributions of Arctic waters, sink as dense water masses to depths of up to 2 km (Mauritzen, 1996; Marshall and Schott, 1999). As an important 'gateway' between the North Atlantic and the Arctic Ocean, the Nordic Seas are therefore one of only few regions globally characterised

by open-ocean convection and deep-water formation, with dense overflow waters leaving the Nordic Seas and constituting the main source of North Atlantic Deep Water (NADW; e.g., Quadfasel and Käse, 2007; Østerhus et al., 2019).

Changes in the Atlantic overturning geometry and -strength were suggested to influence the carbon sequestration efficiency in the Atlantic Ocean, with impacts on atmospheric $CO_2$ ($CO_{2,atm}$) levels (e.g., Yu et al., 2016, 2023; Gottschalk et al., 2019). At present-day, the Nordic Seas are considered a sink for anthropogenic $CO_2$ due to high surface-water $CO_2$ uptake and deep-

water transport, promoted by strong winds, vigorous primary productivity, and deep convection (Sabine et al., 2004; Rysgaard et al., 2009; Watson et al., 2009). During past glacial periods, particularly the last glacial maximum (LGM), overturning in the Nordic Seas was argued to have either weakened (e.g., Yu et al., 2008; Ezat et al., 2014, 2017b, 2021), nearly ceased (e.g., Boyle and Keigwin, 1987; Sarnthein et al., 1994; Rahmstorf, 2002; Thornalley et al., 2015) or remained similar to modern-like conditions (Hoffmann et al., 2013; Oppo et al., 2018; Larkin et al., 2022). However, most of the recent evidence supports

active dense water formation in the Nordic Seas during the LGM (e.g., Ezat et al., 2021; Larkin et al., 2022). In the Atlantic Ocean, NADW is generally believed to have shoaled to above ~2.5 km water depth during the LGM forming Glacial North Atlantic Intermediate Water (GNAIW) (e.g., McManus et al., 2004; Curry and Oppo, 2005; Yu et al., 2008). Outflow of Nordic Seas-sourced water (NSSW) into the North Atlantic during the LGM was suggested to have occurred at water depths above 2.8 km, resulting in high GNAIW carbonate ion concentrations ([$CO_3^{2-}$]; Yu et al., 2008; Ezat et al., 2021), though contributions

to abyssal North Atlantic waters remain debated (Keigwin and Swift, 2017; Ezat et al., 2019; Larkin et al., 2022). Yet, the interplay of Nordic Seas convection and Atlantic overturning via dense overflow water supply and its influence on Atlantic Ocean carbon storage across various timescales remains poorly known.

One geological time interval that serves as an ideal testbed to study the circulation- and carbon cycling dynamics in the Nordic Seas and its impact on the Atlantic Ocean is the climatic optimum of the last interglacial period, i.e. Marine Isotope Stage

(MIS) 5e (130-116 ka before present, BP). MIS 5e is often considered as a potential analogue for future climate changes (e.g.,





Fischer et al., 2018; Guarino et al., 2020), because temperatures in the northern hemisphere were several °C warmer than today (e.g., NGRIP Members, 2004; Clark and Huybers, 2009). However, although this is confirmed by sea surface temperature (SST) reconstructions from the North Atlantic (e.g., Bauch et al., 2012; Rodrigues et al., 2017), reconstructed SSTs in the Nordic Seas were lower than today (e.g., Cortijo et al., 1994; Bauch et al., 1999). In addition, although $CO_{2,atm}$ concentrations

were at pre-industrial levels during MIS 5e (Bereiter et al., 2015), the Arctic Ocean was found to be (nearly) ice-free (Otto-Bliesner et al., 2006; Guarino et al., 2020) during MIS 5e summers, which might have also affected ocean salinities in the Nordic Seas, and hence deep-water formation in that region.

While MIS 5e provides insights into North Atlantic-Nordic Seas dynamics during globally warmer-than-present climate conditions, the subsequent glacial inception and transition into the MIS 4 glacial allow the reconstruction of Nordic Seas deep-

water dynamics (and its potential role in $CO_{2,atm}$ variations) over both glacial-interglacial and millennial timescales. The MIS 5e/d and MIS 5a/4 transitions are both characterised by a decline in $CO_{2,atm}$ levels of ~30-35 ppm (Bereiter et al., 2015) as well as growing northern-hemisphere ice sheets and a drop in global temperatures (e.g., Grant et al., 2012). MIS 4 was linked with an enhanced chemical separation of intermediate and deep water masses in the Atlantic Ocean, facilitating enhanced carbon sequestration in the Atlantic Ocean interior and a drop in $CO_{2,atm}$ concentrations at that time (e.g., Adkins,

2013; Yu et al., 2016). A decline (rise) in bottom water $[CO_3^{2-}]$ and diminished (enhanced) carbonate preservation in Atlantic sediment cores below (above) 3 km water depth (Broecker and Clark, 2001; Broecker et al., 2015; Yu et al., 2016) indicate a similar water mass stratification during MIS 4 and the LGM (Yu et al., 2008; Ezat et al., 2021). However, Nordic Seas overturning dynamics during MIS 4 are insufficiently understood, leaving our understanding of possible impacts on Atlantic overturning and respired carbon accumulation in the Atlantic Ocean fragmentary.

Perturbations in Atlantic overturning and associated northward heat transport were identified during North Atlantic cold phases of the last glacial cycle (i.e., stadials) (e.g., Böhm et al., 2015; Henry et al., 2016). In the Nordic Seas, stadial-interstadial variability was linked with changes in sea ice cover (e.g., Hoff et al., 2016; Sadatzki et al., 2019), in intermediate/surface water temperatures (e.g., Dokken et al., 2013; Sessford et al., 2019), in surface ocean productivity (e.g., Rasmussen et al., 1996; Hoff et al., 2016) and in deep-ocean convection (e.g., Rasmussen et al., 1996; Hoff et al., 2016; Telesinski et al., 2021). Heinrich

stadials (HS), periods of iceberg rafting in the North Atlantic (e.g., Hemming, 2004), are associated with a significant decline in planktonic foraminiferal stable oxygen isotopes in the Nordic Seas (Rasmussen et al., 1996; Bauch and Weinelt, 1997) and an increase in respired carbon storage in the North Atlantic (Yu et al., 2023). However, observational evidence for changes in deep-water formation in the Nordic Seas, and associated variations in carbon storage and overturning strength in the Atlantic Ocean on both glacial-interglacial and millennial-scale (stadial-interstadial) variability is scarce.

Here, we present combined epibenthic foraminiferal (i.e., *Cibicidoides wuellerstorfi*) B/Ca-based bottom water $[CO_3^{2-}]$ reconstructions and -stable oxygen and carbon isotopes in marine sediment core PS1243 (6.533°W, 69.367°N, 2711 m water depth) from the Norwegian Sea, to assess changes in deep convection in the Nordic Seas from MIS 5e to MIS 4. A comparison to a compilation of existing bottom water $[CO_3^{2-}]$ records from the Atlantic Ocean covering the last glacial cycle building on earlier efforts (Yu et al., 2008; Yu et al., 2016) and a complementary abundance record of the aragonitic pteropod *Limacina*





*spp.* from PS1243 sediments provides insights into temporal variations in the corrosivity of bottom waters in the Nordic Seas and the vertical position of the aragonite compensation depth that allows possible links to be drawn between shifts in Nordic Seas- and Atlantic Ocean overturning and -water mass structure.

## 2. Study area

The hydrography of the Nordic Seas is strongly influenced by the interaction between Atlantic Surface Water (ASW) inflowing
from the North Atlantic and Polar Water originating in the Arctic Ocean (Fig. 1). Warm and saline ASW enters the Nordic Seas both east and west of Iceland via the Greenland-Scotland-Ridge (Fig. 1; e.g., Timmermans and Marshall, 2020). The poleward extension of ASW from the North Atlantic Current releases heat to the atmosphere, which leads to surface ocean cooling by up to 7 °C within the Nordic Seas (e.g., Mauritzen et al., 2011; Østerhus et al., 2019). About 70 % (5–6 Sv) of ASW leaves the Nordic Seas after cooling and densification as overflow waters via the Greenland-Scotland-Ridge with
Denmark Strait Overflow Water (DSOW) and Iceland-Scotland Overflow Water (ISOW) contributing equally to the combined volume transport of ~13 Sv south of Greenland (Fig. 1; Dickson and Brown, 1994; Østerhus et al., 2019). South of Greenland, Nordic Seas-sourced overflow waters mix with Labrador Sea Water (6-7 Sv) and continue their southward flow as NADW (~20 Sv) (Fig. 1; Dickson and Brown, 1994; Paillet et al., 1998).

Cold and low-salinity Polar Water from the Arctic Ocean enters the Nordic Seas via the western Fram Strait (Fig. 1; e.g.,
Olafsson et al., 2021). Differences in temperature (T) and salinity (S) between Arctic Ocean- and North Atlantic-sourced waters result in the formation of the Arctic Front (AF, where S=35) and the Polar Front (PF, where S=34.4) (Fig. 1; Swift and Aagaard, 1981) within the Nordic Seas. The convergence of surface water masses between the PF and AF is intrinsically linked with the formation of cold (T=-1–1 °C) and saline (S=34.4–34.9) deep waters (Fig. 1; Swift and Aagaard, 1981; Schäfer et al., 2001; Zweng et al., 2018). Specifically, deep-water convection is triggered by sea ice formation, surface cooling during winter
and dense water supply from the Arctic Ocean resulting from brine rejection on seasonally ice-covered Arctic shelves (Roach et al., 1993; Schott et al., 1993; Marshall and Schott, 1999).

Seawater $[CO_3^{2-}]$ in the central Nordic Seas vary between 120-170 µmol/kg at the surface and 90-100 µmol/kg in bottom waters (Fig. 1b; Lauvset et al., 2022). Deep-water formation in the Nordic Seas results in a weak depth gradient of $[CO_3^{2-}]$ throughout the water column due to the transport of low-$CO_2$ and high-$[CO_3^{2-}]$ surface waters to depth (Yu et al., 2008; Lauvset
et al., 2022). At present-day, young and high-$[CO_3^{2-}]$ deep-water masses from the Nordic Seas are a major component of NADW, which hence has typical $[CO_3^{2-}]$ of >100 µmol/kg (Fig. 1c; Lauvset et al., 2022). Calcite saturation ($\Omega_C$) levels are above 1 everywhere in the Nordic Seas and in the northern North Atlantic (Fig. 1b, c; Lauvset et al., 2022). In contrast, the transition from aragonite over- to undersaturation (where aragonite saturation ($\Omega_A$) equals 1) occurs at ~ 2300 m water depth in the North Atlantic and at ~2000 m in the Nordic Seas (Fig. 1b, c; Lauvset et al., 2022).




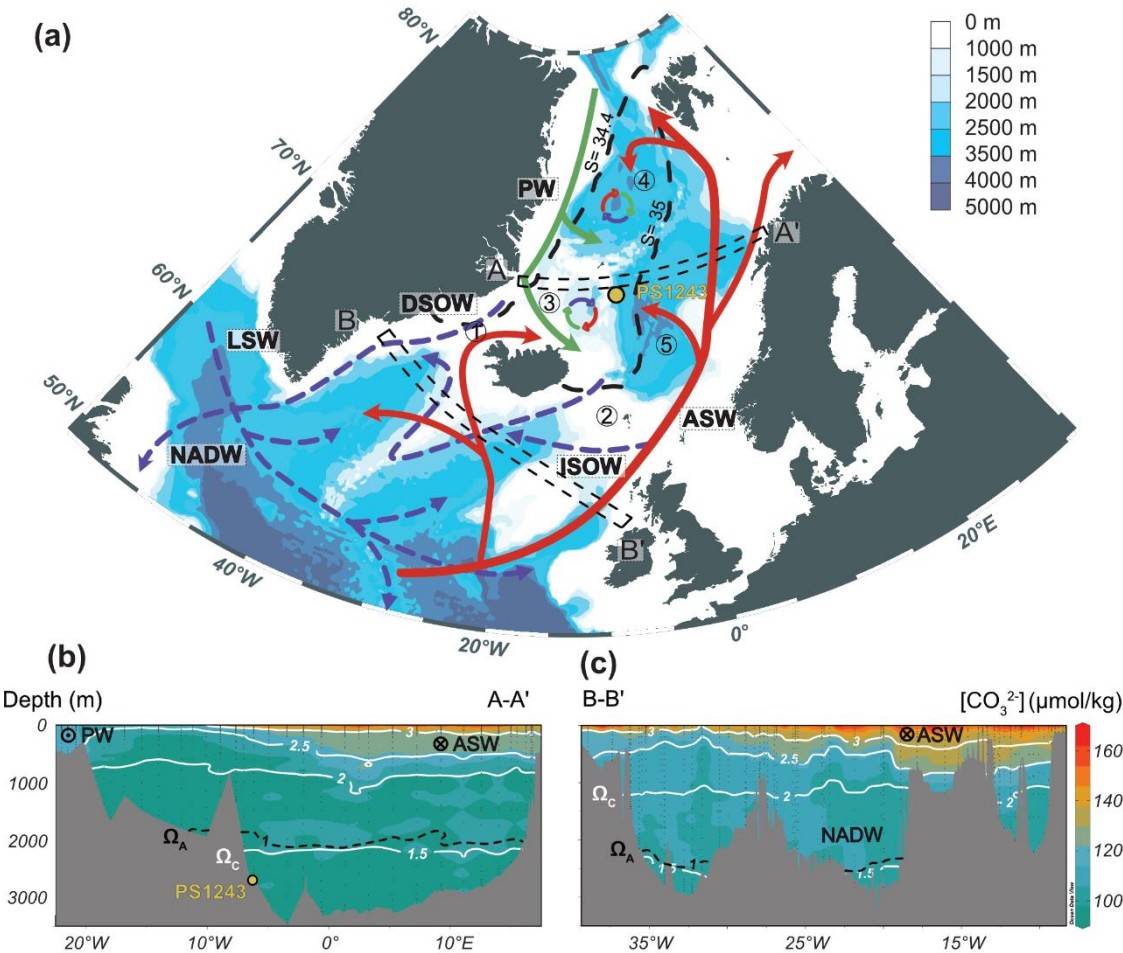

**Fig. 1: Oceanographic setting and carbonate system parameters in the Nordic Seas and North Atlantic. (a) Bathymetry with numbers indicating relevant basins and bathymetric features: 1) Denmark Strait, 2) Iceland-Scotland Ridge, 3) Iceland Sea, 4) Greenland Sea and 5) Norwegian Sea. The arrows show warm- (solid, red) and cold (solid, green) surface currents as well as major deep-water circulation pathways (dashed, blue). Circles with arrows indicate regions of deep convection. Dashed polygons indicate transects shown in (b) and (c). Dashed black lines show isohalines at the surface indicating a salinity (S)=34.4 for the Polar Front and S=35 for the Arctic Front (Zweng et al., 2018). Present-day seawater carbonate ion concentrations ([CO$_3^{2-}$]) in a west-east transect across b) the Nordic Seas (see A-A' in a) and c) the North Atlantic (see B-B' in a) with contours indicating $\Omega_C$ (white, solid contours) and $\Omega_A = 1$ (black, dashed contour). Seawater [CO$_3^{2-}$] are uncorrected for any anthropogenic CO$_2$ uptake. Sections are based on the Global Ocean Data Analysis Project dataset, version 2 (GLODAPv2; Lauvset et al., 2022).**

## 3. Methods

### 3.1 Sediment core PS1243

Sediment core PS1243 was retrieved near the foot of Jan Mayen Ridge in the western Norwegian Basin from a water depth of 2711 m (Fig. 1). The site is located close to the modern AF (Fig. 1). The sediment sequence is a stack of gravity core PS1243-1 and the accompanying giant trigger box core PS1243-2. These were combined using the base of the Vedde Ash layer found in



both sequences (Bauch et al., 2001). Site PS1243 is currently bathed in NSDW (Swift and Koltermann, 1988) and is ideally suited to monitor variations in the deep convection in the Nordic Seas and Nordic Seas-Atlantic Ocean exchange in combination with other sites from the Atlantic Ocean.

Sediment samples from core PS1243 were taken at 1 cm-slices (25 cm³), were freeze-dried and washed over a sieve with a mesh size of 63 µm (Bauch et al., 2001). The coarse fraction was subsequently dried at 45-50 °C to be further processed for stable isotope and trace element analyses on foraminifera as well as pteropod abundance estimates.

### 3.2 Benthic and planktic foraminiferal stable oxygen and carbon isotope analyses

New stable oxygen and carbon isotope measurements were performed on one to five individuals of the epibenthic foraminifer
*C. wuellerstorfi* larger than the 125 µm-fraction from 102 samples at the Alfred Wegener Institute for Polar and Marine Research (AWI) in Bremerhaven (Germany), with a Thermo Finnigan MAT253 isotope ratio mass spectrometer using an automated Kiel carbonate preparation device. Our new data are combined with previously published stable isotope data on *C. wuellerstorfi* performed in core PS1243 (n=127; Bauch et al., 2012). The benthic record is complemented by a continuous isotope record of the planktic foraminifer *Neogloboquadrina pachyderma* (Bauch et al., 2012).

Stable oxygen and carbon isotope analyses are reported in delta-notation ($\delta^{13}C$, $\delta^{18}O$) in ‰ units, referenced against the Vienna Pee Dee Belemnite (VPDB) standard. Long-term reproducibility of new analyses was assessed with an in-house carbonate standard and is <0.04 ‰ for $\delta^{13}C$ and <0.08 ‰ for $\delta^{18}O$. The mean standard deviation of new *C. wuellerstorfi* $\delta^{18}O$ and $\delta^{13}C$ measurements (this study) that replicate published data (Bauch et al., 2012) (n=40) is below 0.06 ‰ VPDB for both $\delta^{13}C$ and $\delta^{18}O$, which is similar to the analytical uncertainties. *C. wuellerstorfi* $\delta^{18}O$ values were corrected by +0.64 ‰ VPDB to account
for disequilibrium effects (Shackleton, 1974; Bauch and Erlenkeuser, 2003).

### 3.3 *C. wuellerstorfi* B/Ca analyses

To measure B/Ca ratios of *C. wuellerstorfi*, 10 to 15 individuals were hand-picked with a dry brush from the sediment fraction larger than 125 µm, resulting in sample weights of 200 to 600 µg $CaCO_3$. In total, 130 samples with sufficient abundance of *C. wuellerstorfi* were analysed. Yet, abundances were insufficient for *C. wuellerstorfi* B/Ca analysis in core sections between
50 – 103 cm and 114 – 138 cm (representing parts of MIS 3 and 2). Given low numbers of *C. wuellerstorfi* tests in 6 samples, consecutive samples were combined over a 2 cm-interval.

Foraminiferal tests were treated according to the oxidative cleaning protocol of Barker et al. (2003) in B-free clean laboratory space. Briefly, each *C. wuellerstorfi* specimen was crushed using a scalpel to ensure opening of all chambers without excessive fragmentation. To remove adhering clay, the samples were rinsed several times with ultra-pure water (18.2 MΩ·cm, Milli-Q
Q-Pod Element) and methanol. To remove organic matter, samples were oxidatively cleaned using alkali-buffered $H_2O_2$. Removal of the coarse silicate grains as described in Barker et al. (2003) was not necessary, as microscopically visible silicates were not observed after the oxidative cleaning step. A weak acid leach with 0.001 M ultrapure $HNO_3$ (prepared from doubly sub-boiled $HNO_3$) was performed to remove any adsorbed contaminants. The samples were subsequently dissolved in 330 µl



0.1 M HNO$_3$ to obtain Ca concentrations via inductively coupled plasma-optical emission spectrometry (ICP-OES, ARCOS

Spectro) at the Institute of Geosciences, Kiel University (Germany). To reduce the surface adsorption of B and thus minimise the memory effect, the samples were diluted with 0.1 M HF (prepared from ROTIPURAN Ultra 48 %) and 0.3 M HNO$_3$ to constant 20 µg/g Ca levels (Misra et al., 2014) and then analysed via inductively coupled plasma-mass spectrometry (ICP-MS, Thermo Agilent 7900) at the Institute of Geosciences, Kiel University (Germany). B/Ca ratios were quantified using a set of gravimetrically prepared calibration standards diluted from single element standards (Alfa Aesar and ThermoFisher Scientific).

All measurements were normalised using the JCt-1 standard (fossil mid-Holocene giant clam *Tridacna gigas,* B/Ca = 217.9 µmol/mol; Inoue et al., 2004).

The analytical B/Ca reproducibility determined by repeat measurements of the same sample aliquot (n=11) is σ = 3.3 µmol/mol. The full procedural reproducibility obtained via true replicate samples (n=7) is σ = 4.3 µmol/mol, which is similar to values reported by Yu and Elderfield (2007).

To quantify past bottom water [CO$_3^{2-}$] at our core site, *C. wuellerstorfi* B/Ca ratios were converted to [CO$_3^{2-}$] following Eq. (1) after Yu and Elderfield (2007):

$$[CO_3^{2-}]_{\text{down-core}} = [CO_3^{2-}]_{\text{pre-industrial}} + (B/Ca_{\text{down-core}} - B/Ca_{\text{core-top}}) / 1.14, \tag{1}$$

Factor 1.14 is the sensitivity of core-top *C. wuellerstorfi* B/Ca ratio variations (in µmol/mol) per µmol/kg change in bottom water [CO$_3^{2-}$] (Yu and Elderfield, 2007). Down-core B/Ca values (B/Ca$_{\text{down-core}}$) refer to *C. wuellerstorfi* B/Ca ratios measured

in PS1243 downcore. The core-top B/Ca value (B/Ca$_{\text{core-top}}$) represents the average B/Ca ratios of our three uppermost samples (2.5-6.5 cm), representing the late Holocene, resulting in B/Ca$_{\text{core-top}}$ = 230.4 ±1.4 µmol/mol (1σ). Pre-industrial [CO$_3^{2-}$] ([CO$_3^{2-}$]$_{\text{pre-industrial}}$) values at the study site are calculated based on modern bottom water [CO$_3^{2-}$] ([CO$_3^{2-}$]$_{\text{modern}}$) corrected for seawater [CO$_3^{2-}$] changes caused by anthropogenic CO$_2$ emissions to the atmosphere. Because Olsen et al. (2010) and Vázquez-Rodríguez et al. (2009) estimated an anthropogenic dissolved inorganic carbon (DIC) addition of 5-10 µmol/kg and

2-21 µmol/kg for the deep Nordic Seas, respectively, we assume an anthropogenic DIC contribution of 10±5 µmol/mol for bottom waters at our study site. Subtracting this value from modern bottom water DIC levels at the study site (station ID5411 (measured 2009): 2166 µmol/kg; Lauvset et al., 2022) and assuming unchanged bottom water alkalinity (station ID5411 (measured 2009): 2302 µmol/kg; Lauvset et al., 2022), estimated bottom water [CO$_3^{2-}$]$_{\text{pre-industrial}}$ at the study site is 101±3 µmol/kg. For the [CO$_3^{2-}$]$_{\text{pre-industrial}}$ calculation, we used the CO2Sys software package (version 2.1) of Pierrot et al.

(2011) with solubility constants K$_1$ and K$_2$ for carbonic acid by Mehrbach et al. (1973), refitted by Dickson and Millero (1987). The dissociation constant for sulfate (K$_{\text{SO4}}$) was used according to Dickson (1990) and the total boron concentration in seawater was calculated after Uppström (1974). In order to assess linkages between deep-water formation and overturning dynamics in the Nordic Seas and the Atlantic Ocean, we compare our new bottom water [CO$_3^{2-}$]$_{\text{down-core}}$ record from the deep Norwegian Sea with similar records from the Atlantic Ocean (Table 1).


**Table 1: Compilation of benthic foraminiferal B/Ca-based bottom water [CO$_3^{2-}$] records from the Atlantic Ocean obtained via laser ablation (LA)- or solution-based inductively coupled plasma mass spectrometry (ICP-MS) analysis. *Core VM28-122 and ODP Site**



999 record intermediate water dynamics of the Atlantic Ocean due to the shallow sill depth at ∼1.8 km water depth of the Caribbean
Sea despite their retrieval from a water depth of 3623 m and 2839 m, respectively.

| Core | Latitude | Longitude | Water depth (m) | Analytical approach | Benthic foraminiferal species | Reference(s) | Age range of B/Ca analyses (ka BP) |
|---|---|---|---|---|---|---|---|
| PS1243 | 69.37°N | 6.55°W | 2711 | ICP-MS | *C. wuellerstorfi* | This study | 1-130 |
| NEAP8K | 59.79°N | 23.90°E | 2360 | ICP-MS | *C. wuellerstorfi* | Yu et al., 2008 | 0-40 |
| BOFS17K | 58.00°N | 16.50°E | 1150 | ICP-MS | *C. mundulus* | Yu et al., 2008 | 0-34 |
| ODP Site 980 | 55.49°N | 14.70°W | 2184 | ICP-MS | *C. wuellerstorfi* | Chalk et al., 2019, Crocker et al., 2016 | 4-145 0-40 |
| BOFS11K | 55.19°N | 20.34°E | 2004 | ICP-MS | *C. wuellerstorfi* | Yu et al., 2008 | 0-36 |
| BOFS10K | 54.67°N | 20.65°E | 2761 | ICP-MS | *C. wuellerstorfi* | Yu et al., 2008 | 0-20 |
| BOFS8K | 52.50°N | 22.06°E | 4045 | ICP-MS | *C. wuellerstorfi* | Yu et al., 2008 | 7-37 |
| BOFS5K | 50.69°N | 21.86°E | 3547 | ICP-MS | *C. wuellerstorfi* | Yu et al., 2008 | 1-32 |
| U1308 | 49.88°N | 24.24°W | 3871 | ICP-MS | *C. wuellerstorfi* | Chalk et al., 2019 | 0-130 |
| U1313 | 41.00°N | 32.96°W | 3426 | ICP-MS | *C. wuellerstorfi* | Chalk et al., 2019 | 5-137 |
| MD95-2039 | 40.58°N | 10.35°W | 3381 | ICP-MS | *C. wuellerstorfi* | Yu et al., 2023 | 0-150 |
| MD01-2446 | 39.38°N | 16.01°W | 3576 | ICP-MS | *C. wuellerstorfi* | Yu et al., 2016 | 51-87 |
| ODP Site 999* | 12.74°N | 78.74°W | 2839* | ICP-MS | *C. wuellerstorfi* | Chalk et al., 2019 | 4-130 |
| VM28-122* | 11.93°N | 78.68°W | 3623* | ICP-MS | *C. wuellerstorfi* | Yu et al., 2010 | 3-160 |
| EW9209-2JPC | 5.60°N | 44.40°W | 3528 | ICP-MS | *C. wuellerstorfi* | Yu et al., 2016 | 50-92 |
| RC16-59 | 4.00°N | 43.00°N | 3520 | ICP-MS | *C. wuellerstorfi* | Broecker et al., 2015 | 0-158 |
| GeoB1115-3 | 3.56°S | 12.56°W | 2945 | LA-ICP-MS | *C. wuellerstorfi* | Raitzsch et al., 2011 | 8-135 |
| GeoB1118-3 | 3.56°S | 16.42°W | 4671 | LA-ICP-MS | *C. wuellerstorfi* | Raitzsch et al., 2011 | 10-135 |
| GeoB1117-2 | 3.81°S | 14.89°W | 3984 | LA-ICP-MS | *C. wuellerstorfi* | Raitzsch et al., 2011 | 8-135 |
| RC13-228 | 22.33°S | 11.20°E | 3204 | ICP-MS | *C. wuellerstorfi* | Yu et al., 2016 | 47-88 |
| RC13-229 | 25.49°S | 11.31°E | 4191 | ICP-MS | *C. wuellerstorfi* | Yu et al., 2016 | 47-101 |
| KNR159-5-78GGC | 27.48°S | 46.33°W | 1820 | ICP-MS | *C. wuellerstorfi* | Lacerra et al., 2017 | 4-25 |
| KNR159-5-33GGC | 27.57°S | 46.18°W | 2082 | ICP-MS | *C. wuellerstorfi* | Lacerra et al., 2017 | 1-20 |
| TN057-21 | 41.13°S | 8.80°E | 4981 | ICP-MS | *C. wuellerstorfi* | Yu et al., 2014 Yu et al., 2016 | 4-28 52-90 |
| MD07-3076 | 44.15°S | 14.23°E | 3770 | ICP-MS | *C. kullenbergi* | Gottschalk et al., 2015 | 0-66 |



### 3.4. Determination of pteropod abundances

We assessed aragonitic shell abundance changes (i.e., weight percentages) of the dominant pelagic pteropod *Limacina spp.* occurring in PS1243 sediments using the 150-500 µm size fraction. Because of the delicacy of pteropod shells, fragments comprise the main portion and were found continuously in every sample between 86 and 189 cm (i.e., from ~31–98 ka). Where available, more complete shells show a clear predominance of the polar cold-water species *Limacina helicina*, along with occasional abundances of the warmer boreal-subarctic species *Limacina retroversa*. Both species are a common part of the polar pelagic fauna today (Bauerfeind et al., 2014). Sediment samples were weighed and split on a glass plate using a razor blade. Shells and fragments were separated from the sample aliquot and weighed with a Sartorius AC121S analytical balance with an accuracy of 0.1 mg in order to calculate the total weight percentage of *Limacina spp.* fragments in the 150-500 µm fraction.

### 3.5. Age model

The age model for the continuously sampled core PS1243 is primarily based on planktic foraminiferal $\delta^{18}O$ compared to the benthic $\delta^{18}O$ stack record (LS16; Lisiecki and Stern, 2016). The stratigraphic alignment was established using a 10 cm-running average of the *N. pachyderma* $\delta^{18}O$ record (Fig. 2a, b). We consider ad-hoc uncertainties for the five resulting age markers of 1.5 ka (Table 2). Millennial-scale (i.e., short-term) variability in the *N. pachyderma* $\delta^{18}O$ record likely results from the influence of low- $\delta^{18}O$ meltwater events during HSs, as commonly assumed for the Nordic Seas (e.g., Rasmussen et al., 1996; Bauch and Weinelt, 1997; Thornalley et al., 2015). To obtain additional age constraints, we calculated a residual *N. pachyderma* $\delta^{18}O$ record (i.e., subtracted the long-term glacial-interglacial $\delta^{18}O$ variability based on the 10 cm-running average). Residual $\delta^{18}O$ minima were then aligned with mid-points of low-$\delta^{18}O$ phases in the NGRIP $\delta^{18}O$ record representing HSs on the AICC2012 ice age scale (Fig. 2). This premise is consistent with the age model for core PS1243 developed for the last 30 ka of Thornalley et al. (2015). To aid identification of HSs recorded via *N. pachyderma* $\delta^{18}O$ minima, we use those *N. pachyderma* $\delta^{18}O$ minima that exceed one standard deviation of the residual record (i.e., those with excursions larger than 0.097 ‰ VPDB) and comprise of at least two data points (Fig. 2). The uncertainty of associated age markers is considered to be half the duration of the respective HS (Table 2; Capron et al., 2021). Our age model also concurs with two northern hemisphere volcanic event marker layers identified in PS1243: the Vedde ash at 44.5 cm (12.1 ka BP; Mortensen et al., 2005; Rasmussen et al., 2006) and an ash layer considered to reflect the North Atlantic Z2 ash at 133 cm (56.1 ka BP; Groen and Storey, 2022). Core PS1243 was shown to resolve millennial-scale variations in Norwegian Sea dynamics despite low sedimentation rates between 1-4 cm/ka (Fig. 2f; e.g., Bauch et al., 2012; Thornalley et al., 2015).





**Fig. 2: Age model for sediment core PS1243 based on a stratigraphic alignment of *N. pachyderma* δ¹⁸O changes to the benthic δ¹⁸O**
**stack (LS16) of Lisiecki and Stern (2016) and water isotope variations (δ¹⁸O) of the Greenland NGRIP ice core (NGRIP Members,**
**2004) on the AICC2012 ice age scale (Veres et al., 2013). (a) *N. pachyderma* δ¹⁸O record of PS1243 and its 10 cm-running average**
**plotted against depth (top axis; in ‰ versus the Vienna Peedee Belemnite (VPDB) standard). (b) LS16 and stratigraphically aligned**
***N. pachyderma* δ¹⁸O in PS1243, based on tie points between the two (open circles and dashed lines, Table 2). (c) Residual *N.***
***pachyderma* δ¹⁸O record of PS1243 (10 cm-running average subtracted). The horizontal dashed lines indicate the 1σ-standard**
**deviation of the residual record of σ=0.097 ‰. Blue lines show established tie points based on LS16 (a, b). Orange areas display short**
***N. pachyderma* δ¹⁸O minima that are interpreted to represent Heinrich stadials (HS1-4, 6, 8-10) and the Younger Dryas (YD),**
**following the rationale of Thornalley et al. (2015); this forms the basis for an additional chronostratigraphic fine-tuning through a**
**stratigraphic alignment of residual *N. pachyderma* δ¹⁸O minima to HSs identified in the NGRIP δ¹⁸O record (Guillevic et al., 2014;**
**Capron et al., 2021) (orange open circles and dashed lines in d, Table 2; see uncertainty assessment in section 3.5). (d) NGRIP δ¹⁸O**
**variations (NGRIP Members, 2004) on the AICC2012 ice age scale (Veres et al., 2013) and the benthic foraminiferal δ¹⁸O stack of**
**LS16 (Lisiecki and Stern, 2016). Light blue bars indicate HSs following the timing given in Capron et al. (2021). e) *N. pachyderma***





$\delta^{18}$O in study core PS1243 on the final age model (combining the stratigraphic alignments in (a) - (d)), and (f) resulting sedimentation rate (SR) changes. (b) – (f) are shown against age (bottom axis). MIS – Marine Isotope Stage.

**Table 2: Tie points of the established age model for sediment core PS1243. The calibrated core-top radiocarbon age is reported in** 260 **Bauch and Weinelt (1997). Tuning targets of *N. pachyderma* $\delta^{18}$O variations in core PS1243 encompass the LS16 benthic foraminiferal $\delta^{18}$O stack from Lisiecki and Stern (2016) and NGRIP $\delta^{18}$O variations (NGRIP Members, 2004 on the AICC2012 ice age scale, Veres et al., 2013). YD – Younger Dryas, HS – Heinrich Stadial, MIS – Marine Isotope Stage**

| Event | Depth (cm) | Age (ka BP) | Error (±ka) | Tuning target |
|---|---|---|---|---|
| Core-top | 1.5 | 1.3 | 0.1 | [14]C dated |
| YD | 45.0 | 12.3 | 0.5 | NGRIP $\delta^{18}$O |
| HS1 | 57.5 | 16.4 | 1.7 | NGRIP $\delta^{18}$O |
| LGM | 61.5 | 19.0 | 1.5 | LS16 |
| HS2 | 70.0 | 25.5 | 2.2 | NGRIP $\delta^{18}$O |
| HS3 | 80.0 | 30.4 | 1.5 | NGRIP $\delta^{18}$O |
| HS4 | 102.5 | 39.2 | 0.8 | NGRIP $\delta^{18}$O |
| HS6 | 143.0 | 61.7 | 2.0 | NGRIP $\delta^{18}$O |
| MIS 4/5 | 163.0 | 71.0 | 1.5 | LS16 |
| HS8 | 176.0 | 85.8 | 1.6 | NGRIP $\delta^{18}$O |
| HS9 | 183.5 | 91.2 | 1.7 | NGRIP $\delta^{18}$O |
| HS10 | 193.0 | 102.5 | 0.7 | NGRIP $\delta^{18}$O |
| MIS 5d | 199.0 | 110.0 | 1.5 | LS16 |
| MIS 5d/e | 209.0 | 117.0 | 1.5 | LS16 |
| MIS 5e/6 | 254.0 | 130.0 | 1.5 | LS16 |

## 4. Results

### 4.1 Efficiency of benthic foraminiferal cleaning for B/Ca analysis

The reconstruction of bottom water conditions based on element/Ca ratios in benthic foraminiferal calcite can be affected by post-depositional contamination and the presence of Mn-carbonate or Mn-Fe oxyhydroxide overgrowths or non-removed clay detritus on or inside foraminiferal tests (e.g., Boyle, 1983; Barker et al., 2003; Misra et al., 2014). To evaluate the efficiency of the applied cleaning for our *C. wuellerstorfi* B/Ca analysis, we assess *C. wuellerstorfi* Al/Ca and Mn/Ca levels and their co-variation with *C. wuellerstorfi* B/Ca ratios in our samples (Fig. 3). *C. wuellerstorfi* Mn/Ca are mostly below 100 μmol/mol

(Fig. 3a), which is considered a benchmark for samples unbiased by contamination (Boyle, 1983). In addition, we do not observe a statistically significant correlation between *C. wuellerstorfi* Mn/Ca and B/Ca values (R=0.01, p=0.9; Fig. 3b) and *C. wuellerstorfi* Al/Ca and B/Ca ratios within 95%-confidence levels (R=0.1, p=0.2; Fig. 3d). Along with the fact that our *C.*





*wuellerstorfi* Al/Ca are consistently low (Fig. 3c) and frequently below the limit of detection (i.e., Al/Ca<11 µmol/mol; n=38), we consider contamination of our B/Ca data to be negligible.

**Fig. 3: Foraminiferal trace element and stable oxygen isotope variations during the last glacial cycle in core PS1243. (a)** *C. wuellerstorfi* **Mn/Ca and (b) cross-plotted against** *C. wuellerstorfi* **B/Ca ratios, (c)** *C. wuellerstorfi* **Al/Ca and (d) cross-plotted**



**against *C. wuellerstorfi* B/Ca ratios. (e) *C. wuellerstorfi* B/Ca ratios, (f) *C. wuellerstorfi* B/Ca-based [CO$_3^{2-}$] reconstructions down-core based on modern and pre-industrial bottom water [CO$_3^{2-}$] values represented by our three core-top values (2.5 – 6.5 cm). (g) *C.***
***wuellerstorfi* δ$^{18}$O and *N. pachyderma* δ$^{18}$O from Bauch et al. (2012) and this study. Boxes at the top indicate marine isotope stages (MIS) according to Lisiecki and Raymo (2005) and marine isotope substages (MIS 5a-5e) after Railsback et al. (2015). Blue bars and numbers indicate Heinrich stadials and the Younger Dryas (YD) following Capron et al. (2021).**

## 4.2. *C. wuellerstorfi* δ$^{13}$C and [CO$_3^{2-}$] variability during MIS 5 and 4

During MIS 5 and 4, *C. wuellerstorfi* B/Ca-derived [CO$_3^{2-}$] at site PS1243 are consistently higher than during the Holocene
(mean [CO$_3^{2-}$]=107±7 µmol/kg, n=28) and [CO$_3^{2-}$]$_{pre-industrial}$ at the core site (101±3 µmol/kg, Fig. 4). Specifically, higher-than-Holocene bottom water [CO$_3^{2-}$] estimates in the deep Norwegian Sea during MIS 5e (117±11 µmol/kg; n=24) are statistically significant within 95% confidence level ($p<0.01$; Fig. 4). In addition, we find a statistically significant correlation between *C. wuellerstorfi* δ$^{13}$C and bottom water [CO$_3^{2-}$] ($R^2$=0.58, $p<0.01$) over the last glacial cycle in the interval covered by our analyses (Fig. 5). High bottom water [CO$_3^{2-}$] characterise MIS 4 (131 ±15 µmol/kg, n=24) and MIS 5b (148 ±9 µmol/kg, n=9),
which coincides with high *C. wuellerstorfi* δ$^{13}$C values during these times (Figs. 4, 5). In other words, both *C. wuellerstorfi* δ$^{13}$C and bottom water [CO$_3^{2-}$] show elevated values during times of low CO$_{2,atm}$ levels (Figs. 4, 5).





**Fig. 4: Multi-proxy bottom water reconstructions in sediment core PS1243 during the last glacial cycle compared to other proxy records. (a)** Water isotope variations (δ¹⁸O) in the Greenland NGRIP ice core (NGRIP Members, 2004) on the AICC2012 ice age
scale (Veres et al., 2013) and atmospheric CO₂ concentrations (Bereiter et al., 2015). **(b)** Bottom water [CO₃²⁻] records of cores PS1243 (this study) and MD95-2039 from the Iberian margin (Yu et al., 2023). **(c)** δ¹³C and **(d)** δ¹⁸O of *C. wuellerstorfi* and *N.*



*pachyderma* in core PS1243. (e) *Limacina spp.* weight percentages in PS1243. Boxes at the top indicate marine isotope stages (MIS) according to Lisiecki and Raymo (2005) and marine isotope substages (MIS 5a-5e) following Railsback et al. (2015). Blue bars and numbers indicate the Younger Dryas and Heinrich stadials following Capron et al. (2021).

Reconstructed bottom water [CO$_3^{2-}$] at site PS1243 increase slightly from MIS 5d to MIS 5c/b, which is paralleled by a similar yet more steady rise in *C. wuellerstorfi* δ$^{13}$C (Fig. 4). The bottom water [CO$_3^{2-}$] increase also coincides with increased weight percentages of *Limacina* spp., both reaching a peak in MIS 5b (Fig. 4). However, a rise in *N. pachyderma* δ$^{13}$C values during MIS 5d precedes the rise in bottom water [CO$_3^{2-}$] during MIS 5c (Fig. 4). Core PS1243 indicates bottom water [CO$_3^{2-}$] variations similar to those in Iberian margin core MD95-2039 during MIS 5e and 5d (Yu et al., 2023), albeit with an average

offset of 22 ±10 µmol/kg (Fig. 4). However, during MIS 5c, coinciding with HS 10, reconstructed bottom water [CO$_3^{2-}$] variability at both sites shows a divergence towards higher (lower) bottom water [CO$_3^{2-}$] values in core PS1243 (MD95-2039) (Fig. 4). At that time, both records begin to anti-correlate on millennial timescales (Fig. 4). Specifically, HS8, 7b and 7a are characterised by high-[CO$_3^{2-}$] bottom waters in Norwegian Sea core PS1243 (this study) and by low-[CO$_3^{2-}$] bottom waters in core MD95-2039 (Fig 4; Yu et al., 2023).

From MIS 5b to 5a, bottom water [CO$_3^{2-}$] in the deep Norwegian Sea slightly decline and increase by ~25 µmol/kg to 160 µmol/kg during MIS 4, when core MD95-2039 indicates a bottom water [CO$_3^{2-}$] minimum of 60 µmol/kg (Fig. 4; Yu et al., 2023). While during MIS 4 *N. pachyderma* δ$^{13}$C values decline and *Limacina* spp. weight percentages decrease to near zero (albeit fragments are still present), *C. wuellerstorfi* δ$^{13}$C values remain high with short-term, low-magnitude minima only (Fig. 4).

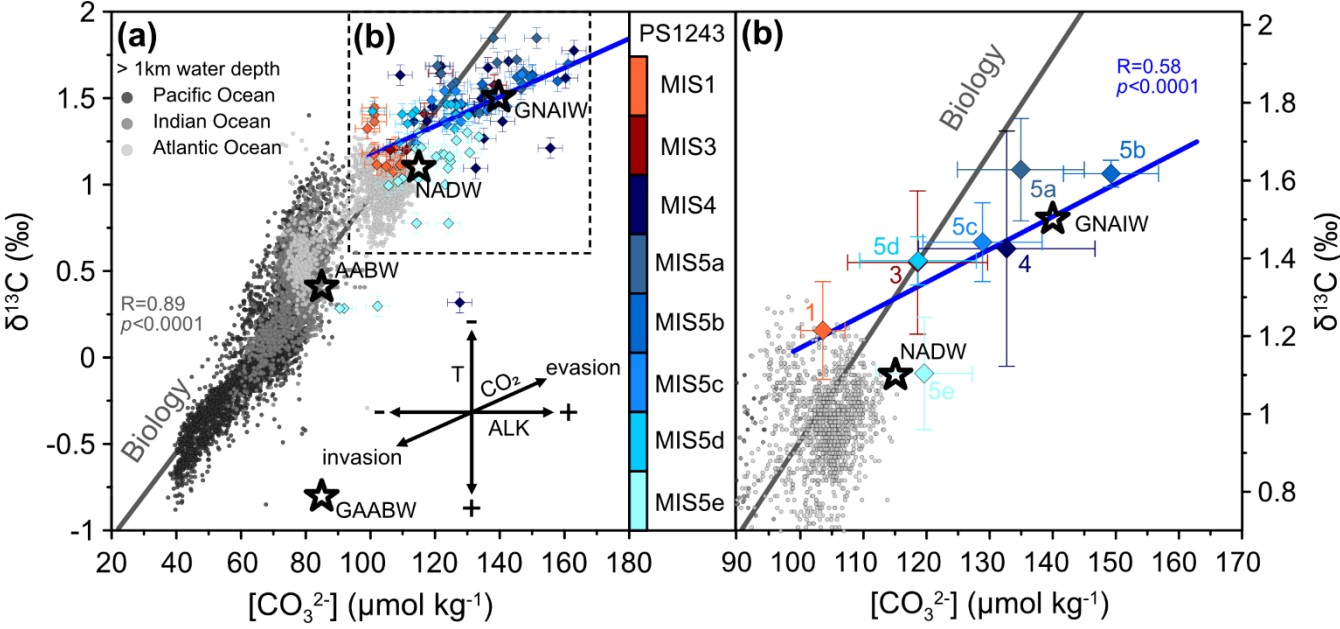


**Fig. 5: Correlation of *C. wuellerstorfi* δ$^{13}$C and B/Ca-based [CO$_3^{2-}$] estimates in sediment core PS1243. (a) Modern seawater δ$^{13}$C and [CO$_3^{2-}$] (Lauvset et al., 2022). Black stars indicate endmember values of North Atlantic Deep Water (NADW), Antarctic Bottom**



Water (AABW), Glacial North Atlantic Intermediate Water (GNAIW) and Glacial Antarctic Bottom Water (GAABW) after Yu et al. (2008; 2020). Diamonds indicate paired *C. wuellerstorfi* $\delta^{13}C$ and B/Ca-based $[CO_3^{2-}]$ estimates in core PS1243 (color-coded

according to time interval) with blue line showing a linear regression of the downcore data (excluding *C. wuellerstorfi* $\delta^{13}C$ values lower than 0.5 ‰). (b) Averaged *C. wuellerstorfi* $\delta^{13}C$ and B/Ca-based $[CO_3^{2-}]$ estimates (stippled box in a) for marine isotope stages (MIS) and substages following Lisiecki and Raymo (2005) and Railsback et al. (2015), respectively (color-coded according time interval), with blue line indicating the linear regression. Grey lines in a) and b) show the Redfield slope, while vectors in the right-bottom corner of panel (a) indicate an increase in seawater $\delta^{13}C$ with decreasing temperature (T), an increase in $[CO_3^{2-}]$ with rising

alkalinity (ALK) and increases in both $[CO_3^{2-}]$ and DIC $\delta^{13}C$ with $CO_2$ evasion/invasion to/from the atmosphere (Yu et al. 2008).

## 4.3. Comparison of bottom water $[CO_3^{2-}]$ records from the Atlantic Ocean and the Norwegian Sea

During MIS 5 and 4, bottom water $[CO_3^{2-}]$ in core PS1243 are consistently higher than reconstructed bottom water $[CO_3^{2-}]$ in the deep Atlantic Ocean below 2.2 km water depth (Table 1), except for Site U1313 whose $[CO_3^{2-}]$ record matches bottom water $[CO_3^{2-}]$ in PS1243 during large parts of MIS 5 (Fig. 6; Chalk et al., 2019). Above 2.2 km water depth in the Atlantic

Ocean, high bottom water $[CO_3^{2-}]$ similar to those recorded in core PS1243 are only observed in sediment cores VM28-122 (~1.8 km water depth; Yu et al., 2010) and at ODP Site 999 in the Caribbean Sea (~1.8 km; Chalk et al., 2019) as well as at ODP Site 980 in the Northeast Atlantic (~2.2 km; Fig. 6c; Crocker et al., 2016; Chalk et al., 2019).

Reconstructed bottom water $[CO_3^{2-}]$ in core PS1243 are statistically indistinguishable (*p*=0.44) during MIS 5a and MIS 4 (Fig. 6). In contrast, all 13 Atlantic sediment cores with $[CO_3^{2-}]$ estimates for the water column below 2.2 km water depth

(Table 1) record a decrease in bottom water $[CO_3^{2-}]$ from MIS 5a to MIS 4 (Fig. 6). This includes sediment cores from the eastern North Atlantic (MD95-2039, MD01-2446, U1308 and U1313), the equatorial Atlantic (RC16-95 and EW9209-2JPC) and the South Atlantic (GeoB115-3, GeoB115-3, GeoB115-3, RC13-229, RC13-228, MD07-3076 and TN057-21) (Table 1). The deep Atlantic sediment cores EW9209-2JPC (~3.5 km water depth; Yu et al., 2016), MD95-2039 (~3.4 km, Yu et al., 2023), RC16-59 (~3.5 km, Broecker et al., 2015) and TN057-21 (~5 km, Yu et al., 2016) with sufficient millennial-scale

resolution indicate low bottom water $[CO_3^{2-}]$ during most stadial conditions of MIS 5 (Fig. 6). This is consistent with persistently low stadial $[CO_3^{2-}]$ in the deep South Atlantic during MIS 3 observed in core MD07-3076 (Gottschalk et al., 2015). In contrast, bottom water $[CO_3^{2-}]$ estimates in PS1243 from the deep Norwegian Sea and in U1313 from the central North Atlantic (Chalk et al., 2019) increase during most stadial periods in MIS 5 (Fig. 6). A divergence of $[CO_3^{2-}]$ on millennial timescales between the deep Atlantic Ocean and the deep Norwegian Sea is particularly highlighted by an anti-correlation of

$[CO_3^{2-}]$ levels in core PS1243 (this study) and Iberian margin core MD95-2039 (Yu et al., 2023) with bottom water $[CO_3^{2-}]$ offsets of 68±11 µmol/kg during HS8, 47±5 µmol/kg during HS7b and 67±5 µmol/kg during HS7a (Figs. 4, 6). In contrast, stadial-interstadial transitions are marked by converging bottom water $[CO_3^{2-}]$ between both cores (Figs. 4, 6).









**Fig. 6: Compilation of Atlantic bottom water [CO$_3^{2-}$] reconstructions based on epibenthic foraminiferal B/Ca for the time interval**
**MIS 5 to MIS 3 (Table 1). (a) Water isotope variations (δ$^{18}$O) in the Greenland NGRIP ice core (NGRIP Members, 2004) on the AICC2012 ice age scale (Veres et al., 2013; grey) and reconstructed bottom water [CO$_3^{2-}$] in core MD95-2039 (Yu et al., 2023; blue). Bottom water [CO$_3^{2-}$] reconstruction obtained in sediment cores (b) PS1243 (this study) and MD95-2039 (Yu et al., 2023), (c) VM28-122 (Yu et al., 2010), ODP Site 999 (Chalk et al., 2019), RC16-59 (Broecker et al., 2015), EW9209-2JPC (Yu et al., 2016), MD01-2446 (Yu et al., 2016), Site U1308 (Chalk et al., 2019), Site U1313 (Chalk et al., 2019) and ODP Site 980 (Chalk et al., 2019; recalibrated**
**based on younger core top B/Ca values of the same core recorded in Crocker et al. (2016)), (d) GeoB115-3, GeoB118-3 and GeoB117-2 (only carbonate ion saturation estimates available; Raitzsch et al., 2011), (e) RC13-229 (Yu et al., 2016) and RC13-288 (Yu et al., 2016), f) MD07-3076 (Gottschalk et al., 2015) and TN057-21 (Yu et al., 2016). The map shows the locations of all cores in the Atlantic Ocean. Boxes at the top indicate marine isotope stages (MIS) following Lisiecki and Raymo (2005) and marine isotope substages (MIS 5a-5e) according to Railsback et al. (2015). Blue bars and numbers indicate Heinrich stadials following Capron et al. (2021).**

## 5. Discussion

### 5.1 Marine Isotope Stage 5e

Despite similar CO$_{2,atm}$ levels (Bereiter et al., 2015) and similar NADW export strength (e.g., Böhm et al., 2015; Chalk et al., 2019) during MIS 5e and the Holocene, bottom water [CO$_3^{2-}$] in Norwegian Sea core PS1243 were slightly, yet significantly higher during MIS 5e than during the Holocene (Figs. 4, 5). In contrast, surface ocean productivity at core site PS1243 was
lower during MIS 5e compared to the Holocene (Thibodeau et al., 2017), which would have resulted in lower surface water (i.e., preformed) [CO$_3^{2-}$] during MIS 5e. Because foraminiferal tests in core PS1243 were slightly better preserved during MIS 5e compared to the Holocene (Henrich et al., 1998, 2002; Helmke and Bauch, 2002), different contributions of alkalinity- and [CO$_3^{2-}$] supply due to CaCO$_3$ dissolution at the bottom can neither explain the observed Holocene-versus-MIS 5e bottom water [CO$_3^{2-}$] offset. In addition, lower-than-Holocene aqueous CO$_2$ (CO$_{2,aq}$) values in the southern Norwegian sea (Ezat et
al., 2017a) at the gateway of inflowing Atlantic surface waters into the Nordic Seas suggest increased preformed [CO$_3^{2-}$] of inflowing North Atlantic (sub-)surface waters that was likely predominantly driven by higher ocean temperatures of North Atlantic surface waters during MIS 5e (i.e. lowered solubility-driven CO$_2$ uptake; e.g., Bauch et al., 2012; Rodrigues et al., 2017). Indeed, *C. wuellerstorfi* δ$^{13}$C- and [CO$_3^{2-}$] values in MIS 5e deviate from Holocene values on a δ$^{13}$C-[CO$_3^{2-}$]-cross plot along vectors indicating higher temperatures and CO$_2$ evasion (Fig. 5; e.g., Yu et al., 2008), supporting a continuous and
efficient transport of the pre-formed surface ocean [CO$_3^{2-}$] signal to depth. We therefore argue that our new bottom water [CO$_3^{2-}$] record reflects continued, if not strengthened, Nordic Seas convection during MIS 5e.

The surface of the Nordic Seas was reconstructed to be colder during MIS 5e than during the Holocene likely due to a reduced heat supply from the Atlantic Ocean to the surface and/or deeper Atlantic water inflow (e.g., Bauch et al., 2012). We argue that along with brine expulsion during an expanded winter sea ice growth growth (Bauch et al., 2012, 2011; Bauch and
Erlenkeuser, 2008) led to persistent convection in the Norwegian Sea and the continuous supply of well-ventilated water masses to depth during MIS 5e. However, additional impacts on Nordic Seas surface ocean buoyancy may have arisen from enhanced summer sea-ice melting in the Arctic Ocean during MIS 5e (Otto-Bliesner et al., 2006, CAPE Last Interglacial Project Members, 2006) and/or increased meltwater supply to the Nordic Seas from MIS 6 ice sheets extending far in MIS 5e (Bauch et al., 2011, 2012), but our new bottom water [CO$_3^{2-}$] data suggest that these did not hamper deep convection in the





Norwegian Sea during that time. Indeed, modelling studies of changes in Nordic Seas overturning forced by various future $CO_{2,atm}$ scenarios suggest a stronger overturning through an increase in the zonal temperature gradient in the Nordic Seas that drives upper-ocean northward transport in the Northeast Atlantic and enhanced deep southward transport near the western boundary in the Northwest Atlantic (Årthun et al., 2023). This was simulated to affect the transformation of surface waters into dense waters and thereby maintain active and strong overturning circulation in the Nordic Seas under warmer-than-present

climate scenarios (Årthun et al., 2023). Considering the global MIS 5e as a warmer-than-present (possible future) climate analogue, our data support these numerical findings and bear witness to a high resilience of Nordic Seas overturning circulation to warmer surface conditions in the North Atlantic (e.g., Bauch and Kandiano, 2007), delayed warming in the Nordic Seas (e.g., Bauch et al., 2011, 2012), changes in surface ocean buoyancy forcing through meltwater supply from surrounding ice sheets (e.g., Bauch et al., 2011) and sea ice reduction in the Arctic Ocean during MIS 5e (e.g., Otto-Bliesner et al., 2006, CAPE

Last Interglacial Project Members, 2006).

Mean bottom water $[CO_3^{2-}]$ during MIS 5e in Norwegian Sea core PS1243 is higher or matches bottom water $[CO_3^{2-}]$ in the North Atlantic Ocean (Fig. 6), supporting the representation of our bottom water $[CO_3^{2-}]$ record as one $[CO_3^{2-}]$ endmember of northern-component waters in the North Atlantic. Our compiled records further suggest that overflows from the Nordic Seas into the eastern North Atlantic (ODP Site 980) persisted during MIS 5e, potentially facilitated by active ISOW (Crocket et al.,

2011). Given a substantial and continuous production of DSOW during MIS 5e (Hillaire-Marcel et al., 2001), despite reductions in Labrador Sea Water formation or even its absence (Cottet-Puinel et al., 2004; Winsor et al., 2012), our data indicate that NSSW remains an important contributor to NADW during this period.

**5.2 Marine Isotope Stage 5e to 5a transition**

Reconstructed bottom water $[CO_3^{2-}]$ in the deep Norwegian Sea remain constant across the MIS 5d/e transition, despite a

~40 ppm decline in $CO_{2,atm}$ (Fig. 4; Bereiter et al., 2015). In addition, $CO_2$ uptake in the upper ocean at the Iceland-Scotland-Ridge increased at that time as evidenced by a rise in $CO_{2,aq}$ concentrations approaching near-equilibrium levels under constant nutrient conditions and decreasing (sub-)SST (Fig. 7; Ezat et al., 2016, 2017a). A $CO_{2,atm}$ drop would operate to increase preformed $[CO_3^{2-}]$ levels in the Nordic Seas, while upper-ocean $CO_2$ uptake and -cooling at the MIS 5e/d transition acts to lower preformed $[CO_3^{2-}]$ levels. The latter is supported by increasing *N. pachyderma* $\delta^{18}O$ levels in core PS1243 and a decline

in reconstructed (sub-)SST in the Norwegian Sea at the MIS 5e/d boundary (Fig. 7; Rasmussen et al., 2003; Ezat et al., 2016). In addition, thermodynamically controlled $\delta^{13}C$ fractionation between the atmosphere and the surface ocean operates to increase DIC $\delta^{13}C$ during cooling (Lynch-Stieglitz et al., 1995), which is reflected in a rise of *N. pachyderma*-, and consequently also *C. wuellerstorfi*, $\delta^{13}C$ levels at our core site (Fig. 7). Based on *C. wuellerstorfi* B/Ca-derived bottom water $[CO_3^{2-}]$ at the Iberian margin, Yu et al. (2023) argued for a strong solubility-driven uptake of $CO_2$ in the North Atlantic and

Nordic Seas because of upper-ocean cooling at the MIS 5e/d transition (i.e., Greenland stadial 26 specifically), which is consistent with our inference from core PS1243. However, we argue that decreasing $CO_{2,atm}$ levels caused a rise in surface water $[CO_3^{2-}]$ in the Atlantic Ocean (i.e., source water $[CO_3^{2-}]$) that acted to raise preformed $[CO_3^{2-}]$ in the study area through



northward transport. This increase in preformed $[CO_3^{2-}]$ was then counterbalanced by cooling-induced increase in $CO_2$ solubility in the Nordic Seas leading to nominally unchanged preformed $[CO_3^{2-}]$ of surface waters at our study site.

Consequently, enhanced deep-water formation, along with potentially minor contributions from changes in sea ice (through $CO_{2,aq}$ enrichment in brines; Rysgaard et al., 2009), surface ocean productivity (influencing surface ocean $[CO_2]$), and carbonate dissolution at the sea floor (releasing alkalinity at depth), maintained constant bottom water $[CO_3^{2-}]$ in the deep Norwegian Sea during the MIS 5d/e transition (Fig. 7). This is consistent with PS1243 *C. wuellerstorfi* $\delta^{13}C$ and $[CO_3^{2-}]$ data during MIS 5e and MIS 5d that are primarily separated from each other in $\delta^{13}C$-$[CO_3^{2-}]$-space along the temperature vector

influencing DIC $\delta^{13}C$ levels only, with different effects on air-sea $CO_2$ exchange cancelling each other out (Fig. 5). Overall, our bottom water $[CO_3^{2-}]$ data suggest continuous deep-water formation in the Nordic Seas at the MIS 5e/d transition, and hence the transformation of lighter upper-ocean waters to dense water masses and their transport to depth in the Norwegian Sea (Figs. 4, 7).

Bottom waters in the deep Norwegian Sea during MIS 5 are generally characterised by higher-than-Holocene $[CO_3^{2-}]$ with a

marked rise starting during HS10 at 103 ka BP (Fig. 6). This suggests overall persistent deep-water formation in the Nordic Seas during that time, which is supported by increased abundances of the pteropod *Limacina spp.* in PS1243 sediments from MIS 5c to 5a (Fig. 4). The appearance of *Limacina* spp. indicates a deepening of the aragonite compensation depth of at least 700 m from its modern position in the Norwegian Sea (~2000 m; Lauvset et al., 2022) and a shift in the aragonite saturation state from present-day levels of $\Omega_A$=0.8 to above 1.0 at core site PS1243 (Fig. 1b). These changes were likely primarily driven

by enhanced or persistent deep Norwegian Sea convection, with possible influences from changes in the export production of *Limacina* spp. shells (Fig. 4), which might have provided a source of alkalinity to bottom waters through (partial) aragonite dissolution, acting to raise bottom water $[CO_3^{2-}]$ (Sulpis et al., 2022). Higher-than-Holocene bottom water $[CO_3^{2-}]$ between 46–40 ka BP and increased abundances of *Limacina* spp. throughout MIS 3 recorded in PS1243 sediments support this notion (Fig. 4). Overall, we argue that despite possible temporal changes in pteropod export fluxes and seafloor aragonite dissolution

rates in the Norwegian Sea over the last glacial cycle that are largely unknown, increased preservation of aragonite shells during MIS 5c to 5a strongly supports the notion of sustained deep-water formation and -ventilation during that time.



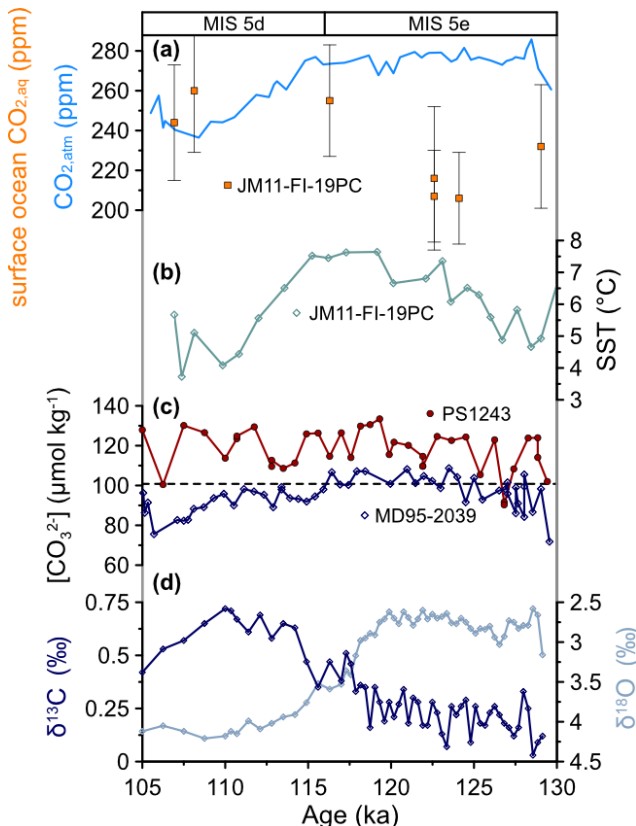

**Fig. 7: Comparison of North Atlantic and Norwegian Sea bottom water [CO₃²⁻] estimates to surface water parameters reconstructed for the MIS 5e-to-5d transition. (a) Atmospheric CO₂ (CO₂,ₐₜₘ) concentration (Bereiter et al., 2015) and reconstructed surface ocean**
**dissolved, aquousCO₂ levels (CO₂,ₐq) corrected by -40 ppm for the calcification depth of *N. pachyderma* in southern Norwegian Sea core JM11-FI-19PC (Ezat et al. (2017a). (b) *N. pachyderma* Mg/Ca-based sea surface temperatures (SST) in core JM11-FI-19PC (Ezat et al., 2016). (c) Bottom water [CO₃²⁻] record of PS1243 (this study) and MD95-2039 (Yu et al., 2023). (d) *N. pachyderma* δ¹⁸O and δ¹³C in core PS1243 (Bauch et al., 2012). Boxes at the top indicate marine isotope stages (MIS) following Lisiecki and Raymo (2005) and marine isotope substages (MIS 5a-5e) according to Railsback et al. (2015).**

**5.3 Transition from Marine Isotope Stage 5a to 4**

At the MIS 5a-to-4 transition, Yu et al. (2016) found a marked decrease in bottom water [CO₃²⁻] by ~25 µmol/kg in Atlantic Ocean cores below 2.8 km water depth (Fig. 6). This, however, contrasts with high bottom water [CO₃²⁻] at ODP Site 999 and in core VM28-122 from the Caribbean Basin at 1.8 km water depth, with ODP Site 980 in the Northeast Atlantic at 2.2 km water depth (Broecker et al., 2015; Chalk et al., 2019) as well as with core PS1243 in the Norwegian Sea at 2.7 km water depth
(Fig. 6). We argue based on our new data that, despite possible changes in preformed [CO₃²⁻] due to lower CO₂,ₐₜₘ levels during MIS 4, the Nordic Seas were characterised by continuous deep convection at that time. In addition, the formation of deep waters in the Nordic Seas and overflows to the Atlantic basin likely contributed to GNAIW that was in turn suggested to be dominantly influenced by a shallow convection cell south of Iceland (Broecker et al., 2015; Chalk et al., 2019). However, the




match of our bottom water [CO$_3^{2-}$] in the deep Norwegian Sea with elevated intermediate Atlantic Ocean [CO$_3^{2-}$] at ODP Sites
980 and 999 and in core VM28-122 (Broecker et al., 2015; Chalk et al., 2019) emphasises strong contributions of well-
ventilated, high-[CO$_3^{2-}$] NSSW to GNAIW via high-density overflows to the intermediate North Atlantic during MIS 4. This
aligns with findings of strong (nearly modern-like) AMOC intensity in the western North Atlantic during MIS 4 (Böhm et al.,
2015) and an increase in the presence of northern-component waters in the Atlantic Ocean at that time (Pöppelmeier et al.,
2021).

The wide-spread decline in deep Atlantic [CO$_3^{2-}$] below 2.8 km water depth was linked with a rise in respired carbon levels
and/or a greater admixture of northward expanded, [CO$_3^{2-}$]-low southern-sourced waters (SSW; Yu et al., 2016). The glacial
northward expansion of SSW in the Atlantic Ocean was postulated to have driven a marked geochemical and density separation
between Glacial Antarctic Bottom Water (GAABW) and GNAIW at 2-3 km water depth (e.g., Adkins, 2013; Yu et al., 2016)
and a concomitant drop in CO$_{2,atm}$ levels through an increase in deep Atlantic respired carbon storage (Yu et al., 2016).

However, while the overflow intensity east of Iceland via ISOW might have decreased during MIS 4, possibly triggered by a
glacial deeper (i.e., subsurface) inflow of ASW (Bauch et al., 2001; Nørgaard-Pedersen et al., 2003; Rasmussen and Thomsen,
2009; Ezat et al., 2021), the expansion of NSSW-influenced GNAIW was likely focussed in the western North Atlantic. This
is consistent with a stronger influence of SSW in the eastern Atlantic basin compared to the Northwest Atlantic Ocean (Chalk
et al., 2019). The contribution of high-[CO$_3^{2-}$] (low-[CO$_{2,aq}$]) NSSW to the Northwest Atlantic would have limited the capacity
of the Atlantic Ocean to store carbon during MIS 4 that likely was mostly stemmed in the eastern basin. Due to the lack of
bottom water [CO$_3^{2-}$] reconstructions below 3 km water depth in the western North Atlantic, contributions of NSSW overflows
to the abyssal Northwest Atlantic during MIS 4, as postulated previously for the LGM (Keigwin and Swift, 2017; Blaser et al.,
2020; Larkin et al., 2022), can neither be identified nor ruled out. Overall, the zonal and meridional extent of high-[CO$_3^{2-}$],
NSSW-influenced GNAIW in the western Atlantic Ocean remains poorly constrained, making a quantification of the impact
of continuous NSSW contributions to the Atlantic and their role in Atlantic Ocean respired carbon levels difficult.

**5.4 Comparison between Marine Isotope Stage 4 and 2**

Our compilation of Atlantic Ocean bottom water [CO$_3^{2-}$] records and our new Norwegian Sea record indicate that the Atlantic
water mass structure and its [CO$_3^{2-}$] characteristics during MIS 2 and MIS 4 share strong similarity, with diverging shallow
Atlantic and deep Atlantic/Nordic Seas [CO$_3^{2-}$] records at the inception of MIS 4 (Figs. 6, 8, Table 1; Yu et al., 2008, 2020;
Broecker et al., 2015; Chalk et al., 2019). We therefore argue that fundamental characteristics of the North Atlantic and the
Nordic Seas were similar during MIS 2 and MIS 4, including a sustained, subsurface inflow of warm, saline Atlantic waters
(Bauch et al., 2001, 2012; Kandiano and Bauch, 2002; Nørgaard-Pedersen et al., 2003; Rasmussen und Thomsen 2009; Ezat
et al., 2014, 2021), continuous overturning circulation and dense water formation in the Nordic Seas (e.g. Veum et al., 1992;
Weinelt et al., 1996; Larkin et al., 2022) and persistent NSSW overflow into the western glacial North Atlantic and marked
contributions to GNAIW and possibly abyssal waters (i.e., glacial NADW (GNADW); Veum et al., 1992; Millo et al., 2006;



Yu et al., 2008; Crocket et al., 2011; Crocker et al., 2016; Ezat et al., 2017b, 2021; Larkin et al., 2022). Additionally, bottom water (i.e., *C. wuellerstorfi*) $\delta^{13}$C and $[CO_3^{2-}]$ during MIS 4 in our Norwegian Sea core PS1243 match, within uncertainties, reconstructed endmember values of GNAIW during MIS 2 ($\delta^{13}$C=1.5‰ VPDB, $[CO_3^{2-}]$=~140 µmol/kg; Fig. 5, Yu et al., 2020). Continued dense water formation in the Nordic Seas during MIS 2 and MIS 4 is also consistent with high *C.*

*wuellerstorfi* $\delta^{13}$C values in core PS1243 (Fig. 4), good, in parts excellent carbonate preservation as supported by our pteropod record in the Nordic Seas during glacials (Henrich et al., 1998, 2002; Helmke and Bauch, 2002) and low subsurface $CO_{2,aq}$ levels in the Norwegian Sea at those times (Ezat et al., 2017a). Furthermore, there is increasing evidence of northern-sourced deep waters in the Atlantic during the LGM (Hoffmann et al., 2013; Howe et al., 2016; Keigwin und Swift 2017; Oppo et al., 2018; Pöppelmeier et al., 2021) likely fed by continuous bottom water sourcing from the Nordic Seas (Blaser et al., 2020;

Larkin et al., 2022). This was also emphasised for MIS 4 by Pöppelmeier et al. (2021) who found a rise in the proportion of northern-component waters in the equatorial and Northeast Atlantic by additional ~15%, which supports our bottom water $[CO_3^{2-}]$-based inferences from core PS1243.

Sustained dense water formation and ventilation of Nordic Seas deep waters during MIS 2 and MIS 4 were likely facilitated by a combination of saline Atlantic water subsurface inflow (Bauch et al., 2001) and a contribution of brines from sea-ice

formation (Veum et al., 1992; Dokken and Jansen, 1999; Bauch and Bauch, 2001). Deep convection was further enforced through the formation of polynyas driven by strong catabatic winds downflowing from proximal ice sheets and shelves (e.g., Bauch et al., 2001; Knies et al., 2018). There remain, however, two contrasting views on the impact of dense NSSW water on the Atlantic Ocean, either as a shallow contribution to GNAIW (e.g., Yu et al., 2008, 2020) or as a dense contribution to abyssal waters resulting in GNADW (e.g., Millo et al., 2006; Howe et al., 2016; Keigwin and Swift, 2017; Ezat et al., 2019).

Irrespective of the exact nature of an NSSW contribution to the North Atlantic Ocean that is ultimately density-dependent, we provide evidence for continuous formation of well ventilated, high-$[CO_3^{2-}]$ NSSW in the Nordic Seas during MIS 4 that likely advected into the northwest Atlantic Ocean via Denmark Strait overflows (Millo et al., 2006; Ezat et al., 2021). Yet, epibenthic foraminiferal B/Ca-derived $[CO_3^{2-}]$ records in the western North Atlantic are limited but urgently needed to estimate the Atlantic Ocean volume influenced by well-ventilated NSSW and to understand the role of the North Atlantic in glacial deep-

sea respired $CO_2$ storage during MIS 2 and MIS 4.





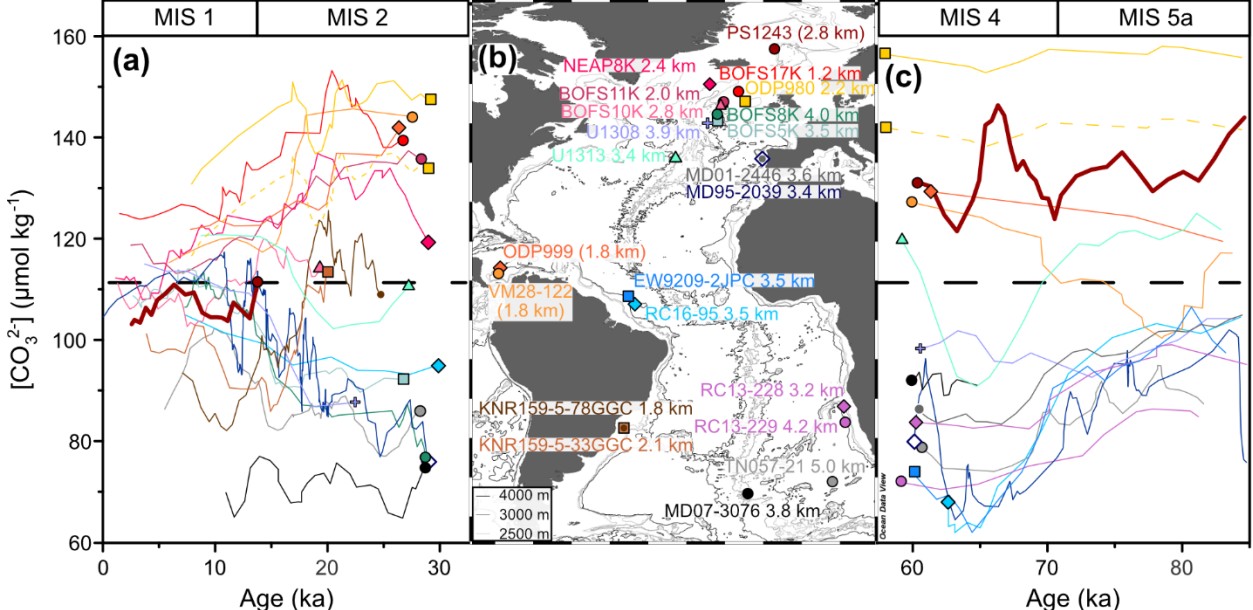

**Fig. 8: Compilation of Atlantic bottom water [CO₃²⁻] reconstructions (five point-running means) for the time intervals a) MIS 2 to MIS 1 (i.e., the last deglaciation) and c) MIS 5a to MIS 4 (i.e., the last glacial inception) (Table 1). Dashed line displays the average reconstructed bottom water [CO₃²⁻] value (~111±26 µmol/mol) from all compiled sediment cores with core-top ages younger than**

**3000 years before present. b) Core locations of epibenthic foraminiferal B/Ca-based [CO₃²⁻] reconstructions from the Atlantic Ocean and the Nordic Seas (Table 1), with symbol type and colour equivalent to those shown at the onset (a) and end (c) of the reconstructed bottom water [CO₃²⁻] records. The bottom water [CO₃²⁻] record of ODP Site 980 (yellow, Chalk et al., 2019) was recalibrated according to younger epibenthic core-top B/Ca values presented in Crocker et al. (2016) (dashed line). Boxes at the top indicate marine isotope stages (MIS) following Lisiecki and Raymo (2005) and marine isotope substages (MIS 5a) according to Railsback et**

**al. (2015).**

## 5.5 Millennial-scale [CO₃²⁻] variability during Marine Isotope Stage 5

Our new bottom water epibenthic B/Ca-based [CO₃²⁻] reconstructions indicate high and/or increasing [CO₃²⁻] values during stadial conditions of MIS 5, specifically HS7a, 7b, 8, 9 and 10 (Fig. 4). This suggests continuously well-ventilated deep-water masses and therefore active convection in the Nordic Seas during these times, contrary to earlier notions (Rasmussen et al.,

1996; Rasmussen and Thomsen 2004; Dokken et al., 2013; Ezat et al., 2014; Hoff et al., 2016; Telesinski et al., 2021). However, our data are in line with the inference of continued stadial deep-water formation due to sea ice formation and brine release of Sessford et al. (2019). Ezat et al. (2017a) found high $CO_{2,aq}$ levels in subsurface waters in the southern Norwegian Sea towards the final stages of the strong and long HS1, HS4 and HS11, which they interpret to indicate suppressed deep-water formation, increased $CO_{2,aq}$ levels (hence low [CO₃²⁻]) of inflowing Atlantic water, decreased productivity and/or $CO_{2,aq}$

enrichment in brines during sea ice formation. Yet, no change in sub-surface $CO_{2,aq}$ levels during HS3 and HS6 was identified (Ezat et al. 2017a), suggesting a delicate balance between processes impacting the carbonate system of the Nordic Seas that may vary among HSs and/or stadial conditions. Therefore, we argue that irrespective of possible preformed [CO₃²⁻] changes of Nordic Seas (sub-)surface waters that likely occurred over millennial timescales, our PS1243 bottom water [CO₃²⁻] record



suggests continued deep convection during stadial conditions of MIS 5, transmitting the (sub-)surface $[CO_3^{2-}]$ signal efficiently to depth. While high-resolution (sub-)surface $CO_{2,aq}$ reconstructions from the Nordic Seas over the last glacial cycle, specifically MIS 5, are required to test our bottom water $[CO_3^{2-}]$-based inferences in core PS1243, our results along with the reconstructions of Ezat et al. (2017a) might signify the regional nature of deep-water formation and preformed $[CO_3^{2-}]$ variations in the Nordic Seas during times of millennial-scale climate variability (e.g., Rasmussen et al., 1996; Ezat et al., 2017a).

Changes in bottom water $[CO_3^{2-}]$ in the deep Norwegian Sea (PS1243, this study) and the deep Iberian Margin site (MD95-2039, Yu et al., 2023) are anti-correlated during MIS 5, with higher $[CO_3^{2-}]$ in the deep Norwegian Sea and lower $[CO_3^{2-}]$ at the Iberian margin during stadials (Figs. 4, 6). Low bottom water $[CO_3^{2-}]$ in core MD95-2039 during these times (Fig. 4) indicate a northward expansion of SSW in the eastern North Atlantic, and therefore enhanced respired carbon due to longer residence times of Atlantic Ocean water masses, weaker ventilation and AMOC slowdown (Yu et al., 2023). In contrast, interstadials are characterised by converging bottom water $[CO_3^{2-}]$ patterns in PS1243 and MD95-2039 (Fig. 4, 6), indicating a stronger contribution of NSSW to Atlantic deep waters at the Iberian margin, in line with AMOC strengthening (Böhm et al., 2015; Henry et al., 2016). Specifically, a shallower inflow of Atlantic surface waters into the Nordic Seas due to less extensive sea ice cover during interstadials (Hoff et al., 2016) might have triggered increased NSSW overflows via the eastern sill with overflows across the western sill remaining active. During stadial conditions, as Atlantic inflow deepened, continued deep-water formation in the Nordic Seas and southern outflow likely via the Denmark Strait mostly, however, would cause zonal gradients in bottom water $[CO_3^{2-}]$ between the deep western and eastern North Atlantic that we argue is depicted in the anticorrelation between bottom water $[CO_3^{2-}]$ in cores PS1243 and MD95-2039 (Figs. 4, 6). In such a scenario, the supply of high-$[CO_3^{2-}]$ (and hence low-$[CO_{2,aq}]$) water masses to the western Atlantic would have acted to diminish the carbon storage capacity of the stadial North Atlantic and partially compensate increased respired carbon storage driven by SSW expansion in the deep eastern Atlantic Ocean (Chalk et al., 2019; Yu et al., 2023). Yet, to unravel the extent of NSSW outflow, its dispersal in the Atlantic Ocean during stadial-interstadial variability and its influence on whole-Atlantic Ocean ventilation and $CO_2$ release during stadials (Yu et al., 2023), bottom water $[CO_3^{2-}]$ reconstructions in the deep western Atlantic are required.

## 6. Conclusion

Using sediment core PS1243 from the deep Norwegian basin, we reconstructed bottom water $[CO_3^{2-}]$ inferred from *C. wuellerstorfi* B/Ca ratios. We have further increased the resolution of the existing *C. wuellerstorfi* $\delta^{13}C$ record for the past 130 ka for the study core and show an unprecedented data set of continuous aragonite preservation in the Norwegian Sea based on the pteropod *Limacina* spp. for much of the last glacial cycle (~31-98 ka BP). The main focus of the study is to assess glacial-interglacial and millennial-scale changes in the overturning strength of the Nordic Seas. An Atlantic Ocean compilation of existing bottom water $[CO_3^{2-}]$ records over the last glacial cycle further allowed a comparison with our new bottom water



[CO$_3^{2-}$] estimates to evaluate potential effects of Nordic Seas convection and dense-water overflows across the Greenland-Scotland Ridge on the chemical water mass structure and respired carbon storage in the Atlantic Ocean.

Bottom water [CO$_3^{2-}$] in the deep Nordic Seas during MIS 5e match published bottom water [CO$_3^{2-}$] records from the Atlantic Ocean, suggesting that our PS1243 [CO$_3^{2-}$] record serves as the [CO$_3^{2-}$] endmember of Nordic Seas-sourced deep waters, and represents one important endmember of northern-component waters in the North Atlantic during that time. The bottom water

[CO$_3^{2-}$] data during MIS 5e further indicate continued deep-water formation in the Norwegian Sea despite a generally warmer North Atlantic and cooler upper Nordic Seas when compared to the Holocene. Such a scenario is consistent with numerical model simulations that attest Nordic Seas overturning a high resilience towards future climate forcing and might therefore testify a stabilising effect of continued Nordic Seas convection and associated overflows on Atlantic overturning (Årthun et al., 2023).

The MIS 5d/e transition was characterised by constant yet high bottom water [CO$_3^{2-}$] in the deep Norwegian Sea. We argue that this primarily reflects persistent deep-water formation in the Nordic Seas and a shift in preformed [CO$_3^{2-}$] levels driven by declining CO$_{2,atm}$ levels that is compensated by CO$_2$ solubility-driven changes in preformed [CO$_3^{2-}$] due to upper-ocean cooling. Higher-than-Holocene [CO$_3^{2-}$] throughout MIS 5 and the dominant occurrence of the pteropod *Limacina* spp. in MIS 5c-to-a sediments indicate a marked deepening of the aragonite compensation depth by at least 700 m in the Norwegian Sea, albeit

effects from changes in pteropod export fluxes cannot be excluded. This bears witness to persistent dense-water formation and deep-water ventilation in the Nordic Seas throughout MIS 5.

The MIS 5a-to-4 transition is marked by a divergence of shallow Atlantic Ocean- and deep Norwegian Sea [CO$_3^{2-}$] from Atlantic [CO$_3^{2-}$] records deeper than 2.8 km water depth, which is reminiscent of similar trends during the LGM (e.g., Yu et al., 2020). This suggests a persistent (subsurface) inflow of Atlantic waters into the Nordic Seas, continuous glacial deep-water

formation in the Nordic Seas and sustained contributions of NSSW to the Atlantic Ocean via dense-water overflows during MIS 4. We argue that this also applies to the more pronounced shorter-term stadial intervals (i.e., within MIS 5), which are also characterised by high [CO$_3^{2-}$] in the deep Norwegian Sea.

Overall, our study puts new constraints on the [CO$_3^{2-}$] end-member composition of deep waters emerging from the Nordic Seas during MIS 5 and MIS 4, and highlights the persistent formation of well-ventilated deep waters in the Norwegian Sea

during these times. However, additional bottom water [CO$_3^{2-}$] reconstructions, primarily from the deep western Atlantic Ocean and the Greenland or Iceland Sea, are needed to further constrain the extent and impact of NSSW on the water mass structure, overturning strength, and ultimately the carbon storage capacity of the Atlantic Ocean throughout the last glacial cycle to fully understand the role of the Atlantic Ocean and the Nordic Seas in Atlantic overturning dynamics and CO$_{2,atm}$ variations.

**Code and data availability:** The map and transects of the study area (Figure 1) were made with Ocean Data View (Schlitzer, 2022) version 5.6.5, available at https://odv.awi.de (last access: 30 July 2024). Data from this study are openly available from the PANGAEA database at [URL to be confirmed] [(citation to be confirmed)].



**Author contributions:** TBS: Conceptualisation, Investigation, Methodology, Visualisation, Writing – original draft
preparation. HAB: Conceptualisation, Resources (stable isotope analysis, samples, pteropods), Writing – reviewing & editing.
DAF: Resources (Laboratory infrastructure), Data curation, Writing – reviewing & editing. JY: Writing – reviewing & editing.
JG: Conceptualisation, Methodology, Visualisation, Funding acquisition (Trace element analysis), Project administration,
Resources (Laboratory infrastructure), Writing – original draft preparation, reviewing & editing.

**Competing interests:** The contact author has declared that none of the authors has any competing interests.

**Acknowledgements:** We thank U. Westernströer and K. Bremer for their assistance with laboratory work, and S. Kraft for
picking and weighing pteropods. We are also grateful to M.M. Ezat for valuable insights into foraminiferal trace element
analyses in the study area.

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
