# Peer review of "Persistent deep-water formation in the Nordic Seas during Marine Isotope Stages 5 and 4 notwithstanding changes in Atlantic overturning"

_EGUsphere, 2024_

## Author Comment (AC1)

**Appendix rebuttal**

[Figure]

Fig. 1

Fig. 2

[Figure]

Fig. 3

Fig. 4

[Figure]

Fig. 5

[Figure]

Fig. 6

Fig. 7

[Figure]

Fig. 8: New Figure

[Figure]

Fig. 9: original Fig. 8

---

## Author Comment (AC2)

**Appendix rebuttal**

[Figure]

Fig. 1

Fig. 2

[Figure]

Fig. 3

Fig. 4

[Figure]

Fig. 5

[Figure]

Fig. 6

Fig. 7

[Figure]

Fig. 8: New Figure

[Figure]

Fig. 9: original Fig. 8

---

## Author Response (AR1)

Stobbe and co-authors present a novel carbonate ion record from the deep Norwegian Sea, spanning the last 130 kyr and encompassing major climate transitions. The record is interrupted during parts of MIS3 and MIS2 and the authors hence focus on interpreting the last glacial inception as well as millennial-scale climate events during MIS5. The carbonate ion record is primarily interpreted as an indicator of past deep convection in the Nordic Seas, with comparisons drawn to other Atlantic records to assess the export of newly formed deep water into the Atlantic basin. However, the latter analysis is somewhat limited due to the absence of carbonate ion records in key regions, preventing a comprehensive assessment of Nordic Seas deep water expansion during the last interglacial period and glacial inception.

The authors provide a commendable and transparent assessment of the inferences drawn from the new record, clearly acknowledging areas of uncertainty and where additional data is required. The manuscript is well-written and illustrated, although some figures may benefit from simplification to enhance clarity, as they are too crowded for my taste.

The authors interpret the new data as indicative of sustained deep water formation not only during the last interglacial period but extending well into the glacial inception, a conclusion that appears to be supported by the data. However, their interpretation of millennial-scale variations in the Norwegian Sea record and attempts to correlate these with Heinrich Stadials are less convincing. Given the (multi-)millennial-scale resolution of the record and significant internal variability, this particular interpretation appears to lack robust support from the data.

I therefore would like to see specifically this issue addressed by the authors before I can recommend publication of the study. Please find more detailed comments of these issues below.

We sincerely thank the reviewer for their valuable feedback, which has greatly contributed to enhancing the quality of the manuscript. We are confident that the changes implemented in response to these suggestions have markedly improved our study.

Detailed responses to each comment are provided below in red. Suggested changes to the main text are italicised. The line numbers refer to the revised manuscript with tracked changes. New references are given at the end of this author response letter.

**Major comment:**

The authors posit that the new PS1243 record resolves millennial-scale events, with particular emphasis on Heinrich Stadials. However, when compared to the reference record MD95-2039 from the Iberian Margin, PS1243 appears to lack the requisite temporal resolution and coherency for reliable interpretation of variability during Heinrich Stadials. Moreover, the carbonate ion record of PS1243 exhibits substantially higher internal noise than MD95-2039, further complicating the identification and interpretation of smaller-scale changes associated with these events. The low sedimentation rate of 1-4 cm/kyr of PS1243 presents a significant constraint on temporal resolution. This limitation is particularly evident during MIS 5c to 4, where the temporal

**sampling frequency appears to be less than one sample/kyr. Such low resolution is consistent with the observed sedimentation rate but raises concerns about the validity of interpreting millennial-scale events. The interpretation of millennial-scale events is a recurring theme throughout the manuscript, featuring in the abstract and receiving detailed treatment in section 5.5. However, given the multi-millennial resolution of the new record, I strongly recommended that the authors exercise caution in interpreting individual data points that coincide with the discussed events. This is particularly crucial considering that the age model may not provide sufficient precision to confidently associate these data points with specific events. In light of these considerations, I advise to reassess the claims regarding millennial-scale event resolution. The authors should consider either refraining from such interpretations or significantly qualifying their assertions, acknowledging the limitations imposed by the record's temporal resolution.**

We acknowledge that interpreting millennial-scale variations in PS1243 is compromised due to low sedimentation rates and the uncertainties of the $[CO_3^{2-}]$ reconstructions. We also agree that we have overemphasised our interpretation of millennial-scale $[CO_3^{2-}]$ in PS1243 in the manuscript without a careful consideration of bioturbational smoothing based on assumptions on bioturbation depths, the abundance of foraminifera and abundance changes of foraminifera as the proxy carriers. These factors ultimately drive the attenuation of proxy signals in marine sediments (e.g., Anderson, 2001; Trauth, 2013).

In response to the valid concerns of both reviewers, we have (i) substantially reduced the emphasis on millennial-scale $[CO_3^{2-}]$ variability of our study core throughout the manuscript (e.g., by significantly shortening the respective paragraph in the introduction). (ii) We remove respective statements from the abstract- and conclusions-sections of the manuscript. (iii) We incorporate reflections on millennial-scale variability in section 5.2 ("Marine Isotope Stage 5e to 5a") rather than in a separate section (previously 5.5 "Millennial-scale $[CO_3^{2-}]$ variability during Marine Isotope Stage 5"). (iv) We add limitations of our interpretations on millennial time scales in the revised manuscript, e.g., in Results section 4.2. and 4.3. and in Discussion section 5.2. For instance, we state "both records [PS1243 and MD95-2039] *tend* to anti-correlate on millennial timescales (Fig. 4), *although the temporal resolution is lower in core PS1243 (Fig. 4)*" (section 4.2; line 575).

We significantly revised the discussion section to be transparent around the limitations of our record, and have included a new figure that addresses bioturbational mixing (new Fig. 8). We state "*At face value, bottom water $[CO_3^{2-}]$ in the deep Norwegian Sea (PS1243, this study) and the deep Iberian Margin site (MD95-2039, Yu et al., 2023) tend to anti-correlate on millennial timescales during MIS 5, with higher $[CO_3^{2-}]$ in the deep Norwegian Sea and lower $[CO_3^{2-}]$ at the Iberian margin during HS (Figs. 4, 6). However, this observation may be biased by bioturbational smoothing of the signal given the low sedimentation rate of our study core, which depends on the mixed layer depth (MLD), abundances of foraminifera (as the proxy carrier) and their variations as well as the magnitude of the proxy signal (e.g., Anderson, 2001; Trauth, 2013). Therefore, we employ sensitivity tests with the simple sediment mixing model of Trauth (2013) to assess the effects of bioturbational mixing on our $[CO_3^{2-}]$ for MLDs ranging between 5 and 15 cm and for a hypothetical (i.e., pre-mixing) 30 µmol kg$^{-1}$-decline (this magnitude matches observations in Yu et al. (2023)) and a hypothetical increase in bottom water $[CO_3^{2-}]$ during HS8 and 9 (Fig. 8). Out of the eight tested scenarios, we find that our proxy-based $[CO_3^{2-}]$ estimates are consistent with a $[CO_3^{2-}]$ increase during HS 8 and 9 (Fig. 8), when overprinted by bioturbation with an MLD of 7 cm that is in fact close to the global mean of ~6 cm (Teal et al., 2008). We hence cannot exclude that the opposite trends between bottom water $[CO_3^{2-}]$ in cores PS1243 and MD95-2039 during HS in MIS 5 are a true signal (Figs. 6, 8).*

*This in turn may hint at continued deep-water formation in the Nordic Seas and a southern outflow in the Atlantic Ocean likely via the Denmark Strait during HS of MIS 5, which could have caused a zonal gradient in bottom water [$CO_3^{2-}$] between the deep western and eastern North Atlantic, and hence between PS1243 and MD95-2039. However, unravelling NSSW [$CO_3^{2-}$] variability, its dispersal in the Atlantic Ocean and its influence on whole-Atlantic Ocean ventilation and $CO_2$ release during stadial-interstadial cycles (Yu et al., 2023) requires high-resolution bottom water [$CO_3^{2-}$] records from high-sedimentation sites in the Nordic Seas as well as from deep Atlantic sites (e.g., from the deep western basin of the Atlantic Ocean).*" (section 5.2; line 884).

**Minor comments:**

**L18: Atmospheric CO2 was increasing by up to 15 ppm during Heinrich Stadials suggesting less not more marine carbon storage as noted here by the authors (even though the terrestrial carbon storage may have also played a role). In general, the community shifted away from attributing all carbon storage changes to water mass changes in the Atlantic, now having a stronger focus on marine carbon storage of the SO and Pacific. Maybe, this can also be reflected here in the Abstract.**

This is corrected. We now acknowledge the role of the Pacific and Southern Ocean in the Abstract: "*Alongside shifts in Pacific- and Southern Ocean carbon cycling*, reductions in the extent and formation of North Atlantic Deep Water (NADW) and the expansion of southern-sourced waters in the Atlantic Ocean were linked to enhanced marine carbon storage during glacial periods […]" (Abstract; line 17).

**L57: Please also cite newer studies, including modelling efforts constrained by proxy data (e.g., Muglia and Schmittner 2021, Pöppelmeier et al., 2023).**

We agree that the study of Muglia and Schmittner (2021) is important and thank the reviewer for the suggestion. We have included it in the following statement flagged by the reviewer: "In the Atlantic Ocean, NADW is generally believed to have shoaled to above ~2.5 km water depth during the LGM forming Glacial North Atlantic Intermediate Water (GNAIW) (e.g., McManus et al., 2004; Curry and Oppo, 2005; Yu et al., 2008; *Muglia and Schmittner, 2021*)."(section 1; line 73). However, we have opted not to include the study of Pöppelmeier et al. (2023) because their observation of a shoaling chemocline of approximately 950 m does not allow for direct attribution to a specific water depth, such as the water depth of 2.5 km that was identified to represent roughly the location of the chemocline during the LGM by our four referenced studies.

**L65: Due to the rapidity of the anthropogenic change there are no real analogues in the past. Maybe this statement should be hence adjusted accordingly.**

We adjust our statement accordingly: "MIS 5e is often considered to provide crucial insights into climate system processes and -feedback mechanisms under future climate conditions (e.g., Fischer et al., 2018; Guarino et al., 2020) […]" (section 1; line 100).

**L71: Please briefly mention the role insolation played in the different conditions of MIS5e vs the Holocene, which explains most of the differences.**

In the revised manuscript, we state the role of changes in summer insolation between MIS 5e and the Holocene more explicitly: "[…] because temperatures in the northern hemisphere were

several °C warmer than today *as a result of a strong orbitally induced maximum in boreal summer insolation* (e.g., NGRIP Members, 2004; Clark and Huybers, 2009; *Past Interglacials Working Group of PAGES, 2015*)." and "*Therefore, fresh water forcing induced by higher Arctic summer insolation during MIS 5e (CAPE Last Interglacial Project Members, 2006)* might have also affected ocean salinities in the Nordic Seas, and hence deep-water formation in that region." (section 1; line 101).

**L229: Please better justify your estimated age uncertainty.**

We correct this age uncertainty to 2.5 kyr, and specify "We consider ad-hoc uncertainties for the five resulting age markers of *2.5 ka (Table 2) considering uncertainties in the alignment itself (~1.5 kyr) and the possibility of regional phase lags in foraminiferal $\delta^{18}O$ (~2 kyr, Lisiecki and Stern, 2016).*" (section 3.5, line 365). We believe that this is a very transparent and conservative assessment of age uncertainties that thanks to the reviewer's comment should provide a better justification for the estimated age uncertainties.

**L231: The additional age constraints of d18O tied to NGRIP is a promising approach, but might also be prone to errors. For instance, there are other instances where the NPS d18O record exceeds the threshold of 1 sigma, but with maybe one data point to little to be identified as an HS. Of course age models are difficult to construct for the Nordic Seas, but can you provide a more thorough assessment of potential age biases due to the employed approach?**

We align residual *N. pachyderma* $\delta^{18}$O minima when they broadly align with HS in the NGRIP record based on our stratigraphic alignment to Lisiecki and Stern (2016) (Fig. 2 a-d). In that respect, this age model approach is a fine-tuning of this first approach, which leads to minor changes in sedimentation rates only (Fig. 2f). We specify this in the revised text "*Using the age model constraints based on our stratigraphic alignment to Lisiecki and Stern (2016),* we identified residual $\delta^{18}$O minima that *broadly coincide with HS recorded in the NGRIP $\delta^{18}O$ record (NGRIP member, 2004; Capron et al., 2021) and with peaks in ice rafted detritus (IRD) supply to PS1243 (Bauch et al., 2012). We then aligned* these with mid-points of low-$\delta^{18}$O phases in the NGRIP $\delta^{18}$O record representing HSs […]" (section 3.5; line 370). If anything, this decreases the age uncertainties of core depths that are far away from tiepoints based on a $\delta^{18}$O alignment to Lisiecki and Stern (2016).

For this age model fine tuning, the attribution of residual *N. pachyderma* $\delta^{18}$O minima to HS in the NGRIP record is based on the fact that these minima exceed the -one-sigma variability of the full record (often reaching the -two-sigma range) \*and\* by the broad match of minima with HS in the NGRIP record based on our stratigraphic $\delta^{18}$O alignment to Lisiecki and Stern (2016). However, "*One exception is HS7a that is close to the MIS 5a/4-tiepoint based on the $\delta^{18}O$ alignment to Lisiecki and Stern (2016) that, if included, might cause large and unrealistic shifts in sedimentation rates*" (section 3.5; line 377).

We are confident in our assumptions regarding the tiepoints based on residual *N. pachyderma* $\delta^{18}$O minima because the link between *N. pachyderma* $\delta^{18}$O minima and HS has been pointed out numerous times previously (e.g., Rasmussen et al., 1996; Bauch and Weinelt, 1997; Bauch et al., 2012; Thornalley et al., 2015), and are mostly accompanied by peaks in IRD (Bauch et al., 2012) (although also subject to bioturbation), as shown in new Fig. 8. Also, we now more carefully consider the age uncertainties of these HS-based tiepoints: "*We consider uncertainties in selecting HS mid-points in the residual N. pachyderma $\delta^{18}O$ record and their equivalents in*

*the NGRIP δ¹⁸O records* to be half the duration of the respective HS in the NGRIP $\delta^{18}$O record *each* (Capron et al., 2021)*, which then propagate to the full age marker uncertainties given in* Table 2." (section 3.5; line 378). Note that this age model approach is consistent with two independently dated tephra layers as highlighted in the text and also in revised Fig. 2 (Appendix), and with earlier age models for the study core reported by Thornalley et al. (2015).

**L242: Does this imply virtually no bioturbation in the core? Any bioturbation should obscure millennial-scale events at this low sed rate.**

A significant impact of bioturbation on the sedimentary record is indeed difficult to exclude at low sedimentation rates, which is why we have carefully adjusted our interpretations of millennial-scale $[CO_3^{2-}]$ variations based on our study core PS1243 (please see our reply to the major comment above). However, the impact of bioturbation on proxy records is determined by a delicate interplay between the sedimentation rate, the bioturbation depth, the abundance of foraminifera in the discrete samples as the proxy carrier (and what fraction ultimately is chosen for the proxy reconstructions), abundance variations of foraminifera across millennial-scale events and the magnitude of the proxy signal (e.g., Anderson, 2001; Trauth, 2013). While we fully agree that bioturbation has attenuated and smoothed millennial-scale variability in our record, in our view it would be hasty to categorically dismiss the possibility of a smoothed representation of millennial-scale variability in our record (of course until otherwise proven). This is due to several reasons: 1) millennial-scale *N. pachyderma* $\delta^{18}$O variability in PS1243 over the last glacial cycle has been linked with (Heinrich) stadials numerous times (e.g., Rasmussen et al., 1996; Bauch and Weinelt, 1997; Bauch et al., 2012; Thornalley et al., 2015), 2) *C. wuellerstorfi* specimens were highly abundant throughout the record, 3) strong abundance variations of *C. wuellerstorfi* are not observed, 4) *C. wuellerstorfi* specimens were very well preserved. Points 2)-4) favour in principle the transmission of a millennial-scale ocean signal into the sediment and its representation in our reconstructions. Nonetheless, we have significantly toned down our interpretations of millennial-scale $[CO_3^{2-}]$ variations in PS1243 and critically assess the limitations of our record regarding smoothing by bioturbation and low sedimentation rates, as highlighted in our reply to the major comment above and based on sensitivity tests with the sediment-mixing model of Trauth (2013) shown in new Fig. 8. We specify in section 3.5 flagged here by the reviewer: "Core PS1243 was shown to resolve millennial-scale variations in *surface ocean hydrography* in the Norwegian Sea despite low sedimentation rates between 1-4 cm/ka, *which likely results from high foraminiferal abundances, excellent carbonate preservation and/or low bioturbation depths* (Fig. 2*c*-f; e.g., Bauch et al., 2012; Thornalley et al., 2015)"(line 384).

**L286: Maybe I misunderstand, but 107 +- 7 seems to be well within error of 117 +- 11 µmol/kg for the Holocene and MIS5e. Also, are these 1 sigma uncertainties?**

As explained in the manuscript, we conducted a two-sample student *t*-test, which demonstrated that the differences between the interglacial averages are statistically significant at the 95% confidence level. This finding indicates that in only 5 out of 100 cases the samples with the given mean, 1 sigma-standard deviation and sample size would indicate the same "state", i.e., the sample population (confidence level of 95%). Even when including the full uncertainties of our reconstructions (summing to ±13 µmol/kg for the Holocene and ±15 µmol/kg for MIS 5e, which is now corrected in the manuscript) and increasing the significance level to 10%, this outcome remains robust (i.e., *p*<0.005). This statistically significant difference in $[CO_3^{2-}]$ aligns with suggested differences in surface ocean productivity (Thibodeau et al., 2017), in carbonate preservation (Henrich et al., 1998, 2002; Helmke and Bauch, 2002), in surface ocean temperature (Bauch et al., 2012) at the core site as well as in the preformed $[CO_3^{2-}]$ signature

of the source waters (Ezat et al., 2017b), as highlighted in the manuscript (section 5.1). In addition, they coincide with statistically significant differences (two-sided student $t$, p<<0.025) in *C. wuellerstorfi* $\delta^{13}C$ between the Holocene ($\delta^{13}C$=1.29±0.12 (1σ), n=19) and MIS 5e ($\delta^{13}C$=1.09±0.20 (1σ), n=46). Combined these findings suggest slight yet significant differences in Nordic Seas dynamics/hydrography that likely also affected the degree/rate of deep convection in that region which we discuss in the study (section 5.1). We therefore opt to stick to our interpretations of Holocene-MIS5e-[$CO_3^{2-}$] differences, supported by the statistics, yet point out that the Holocene-MIS5e-difference is only *slight* (throughout section 5.1) and that these finding beg to be crossed-checked with other cores from the Nordic Seas and/or other proxy analyses ("*However*, additional bottom water [$CO_3^{2-}$] reconstructions, primarily from the deep western Atlantic Ocean and the Greenland or Iceland Sea, are needed to further constrain *the nature of Nordic Seas overturning* […]", Conclusions).

**L307: There does not seem to be a clear millennial-scale variability in PS1243 as is seen in the Iberian Margin record. The statement of the anti-correlation therefore seems not fully supported by the data.**

This is an important caveat of our timeseries that we have not sufficiently acknowledged in the manuscript. In response to the reviewer's major comment and on L. 242 (see above), we address this issue more carefully in the revised manuscript. We state "At that time, both records *tend* to anti-correlate on millennial timescales (Fig. 4), *although the temporal resolution is lower in core PS1243 (Fig. 4)*" (section 4.2; line 575). Please refer to further details in our reply to the other comments on this topic, and our revisions to the Discussion section of the manuscript.

**L335: This is really hard to see in Fig. 6 as so many lines overlap. Can you visualize this more clearly? Further, the GeoB records have a very low temporal resolution, which makes such statements not supported by the data for these records.**

We recognise that visualising multiple datasets is challenging, and that our previous Fig. 6 was sub-optimal in some respects. We have revised Fig. 6 accordingly: (i) We rearrange time series according to their geographic location and separate them on different axes (e.g., North Atlantic, tropical Atlantic and South Atlantic), (ii) we convert Δ[$CO_3^{2-}$] to [$CO_3^{2-}$] for the GeoB cores of Raitzsch et al. (2011) following Yu et al. (2016). Here we opt to include these sites as the observed decrease in bottom water Δ[$CO_3^{2-}$] is a robust signal despite the low temporal resolution of the GeoB records, and the decrease is consistently observed in three independent sediment cores – a statement that was also highlighted and confirmed in Yu et al. (2016). (iii) Lastly, we have removed full core names from the map and replaced them with core numbers (referencing to full metadata information in Table 1) for a better overview of core sites. Symbols in the map are identical to those of the timeseries.

**L342: U1313 (and also PS1243) does not really have the temporal resolution to make such statements. Mostly, just a single data point falls into the stadial periods.**

In light of our response to the reviewer's comments on the possible limitations of our study core in documenting millennial-scale changes (see above), we have significantly toned down this statement, and discuss potential influences from bioturbation on the record. It now reads: "In contrast, bottom water [$CO_3^{2-}$] estimates in PS1243 from the deep Norwegian Sea *do not show a [$CO_3^{2-}$] decline during stadial periods but rather a slight increase before or during HS periods* in MIS 5 *(Fig. 4, 6)*" (section 4.3; line 628). We discuss the limitation of this observation thereafter: "*Given the low sedimentation rate of our study core, the uncertainties of our bottom water [$CO_3^{2-}$] estimates and the smoothing effect of bioturbational overprints,*

*we consider this to point at either an attenuated positive HS [CO$_3^{2-}$] anomaly, the lack of an HS [CO$_3^{2-}$] anomaly or the complete removal of a negative HS [CO$_3^{2-}$] anomaly via strong smoothing.*" (section 4.3; line 630).

**L366: As mentioned before, the average CO32- concentrations at MIS5e and the Holocene agree within error. The discussion should therefore be more nuanced on this regard.**

Please refer to our response to the reviewer's comment on line 286 (above), where this point is addressed in detail

**L375: Galaasen et al. (2020, Science) suggested centennial to millennial scale NADW variability also during MIS5e. Do you see any such variability, or rather could such events add to some noise the PS1243 record? Or can you exclude these events?**

The study of Galaasen et al. (2014), where the MIS 5e data were first published, is indeed an interesting study to consider in the context of our data. Although spatial and temporal dynamics in North Atlantic deep-water formation at the Eirik Drift south of Iceland and in the Nordic Seas north of Iceland likely differed, a bottom water [CO$_3^{2-}$] decline of ~25 µmol/kg at 127 ka observed in our record could indeed reflect a short-term perturbation in deep-water formation at our study site equivalent to what is seen south of Iceland. This interpretation is supported by the co-occurrence of *C. wuellerstorfi* and *N. pachyderma* δ¹⁸O minima and the supply of ice-rafted detritus in core PS1243 at that time (Bauch et al., 2012), which are likely linked to the supply of significant amounts of meltwater, similar to patterns observed during the last deglaciation by Thornalley et al. (2015). This also matches new observations for early MIS 5e in Ezat et al. (2024). However, the overall high bottom water [CO$_3^{2-}$] values during MIS 5e suggest that a possible related perturbation in Nordic Seas overturning was short-lived and/or did not affect MIS5e strongly beyond centuries or (sub-)millennia, especially after 126 ka before present. We acknowledge this in the revised manuscript accordingly: "However, additional impacts on Nordic Seas surface ocean buoyancy may have arisen *from the import of Arctic buoyancy anomalies* (*likely related to* enhanced summer sea-ice melting in the Arctic Ocean) during MIS 5e (Otto-Bliesner et al., 2006, CAPE Last Interglacial Project Members, 2006; *Ezat et al., 2024*). There is also multiple evidence for prolonged meltwater supply from remnant MIS 6 ice sheets surrounding the Nordic Seas which extended far into MIS 5e (Bauch et al., 2011, 2012) causing a late thermal maximum in the upper ocean (Bauch et al., 2011; Capron et al., 2014; Zhuravleva et al 2017) that may have been linked with centennial- to sub-millennial-scale disruptions in NADW formation (e.g. Galaasen et al., 2014).* Our new bottom water [CO$_3^{2-}$] data suggest that *these additional influences on Nordic Seas surface ocean buoyancy* did not hamper deep convection in the Norwegian Sea *beyond centuries or few (sub-)millennia.*" (section 5.1; line 723).

While the presence of this short-term perturbation in convection supports the interpretation of centennial/millennial variability represented in our low-sedimentation environment (see our reply to the major comment), we refrain from discussing these centennial-scale or sub-millennial-scale perturbations in detail, for several reasons: (i) the event of enhanced meltwater supply in PS1243 during early MIS 5e/end of Termination II were discussed in another publication that we cite (Bauch et al. 2012) and (ii) there is still a large debate around the mechanisms driving this event, either meltwater supply from remnant glacial ice sheets surrounding the North Atlantic (Hodell et al. 2009; Bauch et al., 2012), increased freshwater supply from the Arctic Ocean (Ezat et al., 2024) and/or deglaciation of the Greenland ice sheet (Galaassen et al. 2014, 2020; Zhuravleva et al., 2017). Age models and their consistency across cores become crucial in this discussion. We believe that covering these details go beyond the

scope of our paper. However, we hope that citing relevant studies and mentioning possible perturbations on shorter timescales sufficiently covers these ambiguities in the revised manuscript.

**L379: "growth growth" delete one.**

Corrected.

**L390: Most models than run beyond 2100 show eventually an AMOC recovery often to a stronger state than under PI conditions. See for instance the results of the LongRunMIP (Bonan et al., 2022). Only transiently the AMOC and Nordic Seas deep water formation weakens. On the timescale this study looks at, one would expect a stronger than Holocene circulation at MIS5e.**

Thank you for mentioning this study that supports our findings. We include details of the study of Bonan et al. (2022), along with the previous model study (Årthun et al., 2023), in the revised manuscript: "*Indeed, modelling studies forced by various future CO$_{2,atm}$ scenarios suggest a weakening of the AMOC during the first century and a decline in overturning in the Nordic Seas until ~2050 that is followed by an AMOC recovery to the initial or stronger state (depending on the model; Bonan et al., 2022) and a strengthening of overturning in the Nordic Seas thereafter (Årthun et al., 2023), respectively. This is consistent with our observation of persistent overturning in the Norwegian Sea during MIS 5e that might have been slightly stronger than during the Holocene. The simulations suggest that stronger salinity- and temperature gradients between high- and mid-latitudes* affected the transformation of surface waters into dense waters and thereby maintain active and strong overturning circulation both in the Nordic Seas and the Atlantic Ocean under warmer-than-present climate scenarios (*Bonan et al., 2022*; Årthun et al., 2023). Considering the global MIS 5e as a warmer-than-present (possible future) climate scenario, our data bear witness to a high resilience and/or fast recovery of Nordic Seas overturning circulation to […] *that likely had a stabilising effect on the AMOC overall (Bonan et al., 2022; Årthun et al., 2023).*" (section 5.1; line 783). And "The bottom water [CO$_3^{2-}$] data during MIS 5e further indicate overall persistent deep-water formation in the Norwegian Sea, *despite the possibility of centennial- or (sub-)millenial-scale phases of weakening* […]. Such a scenario is consistent with numerical model simulations that attest *overturning in the Nordic Seas and the Atlantic Ocean* a high resilience towards future climate forcing *on longer timescales* and might therefore testify a stabilising effect of continued Nordic Seas convection and associated overflows on Atlantic overturning (*Bonan et al., 2022;* Årthun et al., 2023)." (Conclusions; line 1068).

**L430: After HS10 not during.**

Corrected.

**L483: Maybe phrase more carefully, since PS1243 does not exhibit a MIS2 section.**

This is adjusted to "*Although core PS1243 lacks bottom water [CO$_3^{2-}$] estimates for MIS 2,* our compilation of Atlantic Ocean bottom water [CO$_3^{2-}$] records  indicate that the Atlantic water mass structure and its [CO$_3^{2-}$] characteristics during MIS 2 and MIS 4 share strong similarity, with diverging shallow Atlantic and deep Atlantic/Nordic Seas [CO$_3^{2-}$] records at the inception of MIS 4 […]." (line 983)

**L505: promoted instead of enforced.**

Corrected.

**L527 following: As mentioned before, the record of PS1243 does not really have the resolution to resolve these millennial-scale events. The following paragraph therefore seems too speculative and not well-enough supported by the data.**

Please refer to our detailed response to the reviewer's major comment regarding this important topic and related changes to the manuscript.

**Fig. 1A: Please highlight the core location a bit more predominantly. The panel generally feels a bit too busy. Maybe some elements can be removed or highlighted differently (e.g., there are a lot of dashed lines).**

We revised Fig. 1a by (i) removing redundant circulation arrows, (ii) showing oceanic fronts as stippled lines and bottom water currents as solid lines, and (iii) by increasing the symbol size of the core location symbol to make it more prominent.

**Fig 3: The Mn/Ca ratio seems to have a consistent downward trend from 130 ka to 70 ka. How can this be explained?**

Manganese precipitates in sediments under oxic and/or reducing conditions and can accumulate in authigenic coatings around foraminiferal test as Mn-Fe-oxyhydroxides and/or Mn-carbonates (e.g., Boyle, 1983; Hasenfratz et al. 2017; Detlef et al., 2020; Öğretmen et al., 2022). However, benthic foraminifera can also incorporate Mn into their pristine carbonate tests upon biomineralisation, likely most prevalent in conditions of low bottom water oxygen conditions, high $Mn^{2+}$ availability in foraminiferal microhabitats and/or elevated export production (e.g., Koho et al., 2007; Groeneveld and Filipsson, 2013; Öğretmen et al., 2022). These systematics are complex and are insufficiently understood for *C. wuellerstorfi*. While the first may impact measured B/Ca levels, the latter does likely not. Understanding the systematics of our *C. wuellerstorfi* Mn/Ca record goes, however, beyond the scope of our study, and we have therefore refrained from adding explanations on this very topic in the text.

The importance of the *C. wuellerstorfi* Mn/Ca record for our study lies in the impact of authigenic Mn-phases on *C. wuellerstorfi* B/Ca as they may also incorporate B. We can exclude a major impact of contaminant coating phases on our *C. wuellerstorfi* B/Ca record, because *C. wuellerstorfi* Mn/Ca levels are generally lower than 100 μmol/mol that is used as the standard limit to identify clean foraminiferal tests (i.e., those unbiased by contamination; Boyle, 1983) and there is no significant correlation between *C. wuellerstorfi* Mn/Ca and B/Ca ratios (R=0.01, *p*=0.9), as explained in section 4.1.

Even if we consider that most of the Mn/Ca signal is caused by a B-rich extraneous phase (which would be an overestimation given some likely incorporation of Mn into the test) that has a maximum concentration of 350 ppm equivalent to what is found in Fe-Mn nodules (which is likely an overestimation too; Baturin, 1988), then the calculated contaminant B/Ca levels in our study core are negligibly small. Specifically, the total amount of Mn in the samples can be derived from the weight of our samples after cleaning (~300 μg) and measured Mn/Ca ratios of the individual samples. Taking a concentration of 20% and 0.035% for Mn and B in the contaminant phase respectively into account (equivalent to Fe-Mn nodules; Baturin, 1988), the calculated maximum contaminant B/Ca levels in our foraminiferal samples is 0.4 μmol/mol. This is less than 0.2% of mean B/Ca levels in core PS1243 (~220 μmol/mol) and reinforces our

argumentation in section 4.1. of a negligible bias of Mn-rich coatings on our benthic foraminiferal B/Ca record, and hence our interpretations.

**Fig. 8: All three panels are very crowded and it's hard to get a good overview of the records and core sites. I don't have an obvious suggestion to redesign the figure, but I would greatly appreciate if the authors find a cleaner way to visualize the data.**

We are aware of the crowded nature of this figure and have implemented changes to improve its clarity: (i) we illustrate core locations in two separate maps depending on the water depth of the core site (above and below 3 km) as this also separates general trends in seawater $[CO_3^{2-}]$ changes. (ii) We replace the rather long core labels in the map with numbers (that cross-reference to the metadata of the core sites in existing Table 1). (iii) We increase x-axes lengths and place the deep (>3 km) and shallow cores (<3 km) on different y-axes to enhance clarity. (iv) We equally scale the x-axes. (v) We use a colour coding from north (warm colours) to south (cold colours) for time series and core symbols. We hope that these changes sufficiently enhance the clarity of the figure.

Nonetheless, we also want to emphasise that in revised Fig. 9 (originally Fig. 8) it is not the detail of a single time series or the relationship of every individual time series towards each other that is important. Rather, the main message are the opposing trends in $[CO_3^{2-}]$ changes in cores identified shallower or deeper than 3 km water depth, which we hope becomes sufficiently clear from revised Fig. 8 (now Fig. 9).

References:

Anderson, D. M.: Attenuation of Millennial-Scale Events by Bioturbation in Marine Sediments, Paleoceanography, 16, 352–357, https://doi.org/10.1029/2000PA000530, 2001.

Baturin, G. N.: The Geochemistry of Manganese and Manganese Nodules in the Ocean, 342, D. Reidel Publishing Company, Dordrecht, Holland, 1988.

Bauch, H. A. and Weinelt, M. S.: Surface water changes in the Norwegian sea during last deglacial and holocene times, Quaternary Sci. Rev., 16, 1115–1124, https://doi.org/10.1016/S0277-3791(96)00075-3, 1997.

Bauch, H. A., Kandiano, E. S., and Helmke, J. P.: Contrasting ocean changes between the subpolar and polar North Atlantic during the past 135 ka, Geophys. Res. Lett., 39, https://doi.org/10.1029/2012GL051800, 2012.

Bauch, H. A., Kandiano, E.S., Helmke, J.P., Andersen, N., Rosell-Mele, A., and Erlenkeuser, H.: Climatic bisection of the last interglacial warm period in the Polar North Atlantic, Quaternary Sci. Rev., 30, 1813–1818, https://doi.org/10.1016/j.quascirev.2011.05.012, 2011.

Boyle, E. A.: Manganese carbonate overgrowths on foraminifera tests, Geochim. Cosmochim. Ac., 47, 1815–1819, https://doi.org/10.1016/0016-7037(83)90029-7, 1983.

CAPE-Last Interglacial Project Members: Last Interglacial Arctic warmth confirms polar amplification of climate change, Quaternary. Sci. Rev., 25, 1383–1400, https://doi.org/10.1016/j.quascirev.2006.01.033, 2006.

Capron, E., Govin, A., Stone, E. J., Masson-Delmotte, V., Mulitza, S., Otto-Bliesner, B., Rasmussen, T. L., Sime, L. C., Wael- broeck, C., and Wolff, E. W.: Temporal and spatial struc- ture of multi-millennial temperature

changes at high latitudes during the Last Interglacial, Quat. Sci. Rev., 103, 116–133, https://doi.org/10.1016/j.quascirev.2014.08.018, 2014.

Detlef, H., Sosdian, S.M., Kender, S., Lear, C.H., and Hall, I.R.: Multi-elemental composition of authigenic carbonates in benthic foraminifera from the eastern Bering Sea continental margin (International Ocean Discovery Program Site U1343), Geochim. Cosmochim. Acta, 268, 1-21, https://doi.org/10.1016/j.gca.2019.09.025, 2020.

Ezat, M. M., Rasmussen, T. L., Hönisch, B., Groeneveld, J., and deMenocal, P.: Episodic release of $CO_2$ from the high-latitude North Atlantic Ocean during the last 135 kyr, Nature Commun., 8, 14498, https://doi.org/10.1038/ncomms14498, 2017b.

Ezat, M. M., Fahl, K., Rasmussen, T.L.: Arctic freshwater outflow supressed Nordic Seas overturning and oceanic heat transport during the Last Interglacial, Nature Commun., 15, 8998, https://doi.org/10.1038/s41467-024-53401-3, 2024.

Galaasen, E. V., Ninnemann, U. S., Irvalı, N., Kleiven, H. K. F., Rosenthal, Y., Kissel, C., and Hodell, D. A.: Rapid reductions in North Atlantic Deep Water during the peak of the last interglacial period, Science, 343, 1129–1132, https://doi.org/10.1126/science.1248667, 2014.

Galaasen, E. V., Ninnemann, U. S., Kessler, A., Irvalı, N., Rosenthal, Y., Tjiputra, J., Bouttes, N., Roche, D. M., Kleiven, H. K. F., and Hodell, D. A.: Interglacial instability of North Atlantic deep water ventilation, Science, 367, 1485–1489, https://doi.org/10.1126/science.aay6381, 2020.

Groeneveld, J. and Filipsson, H. L.: Mg/Ca and Mn/Ca ratios in benthic foraminifera: The potential to reconstruct past variations in temperature and hypoxia in shelf regions, Biogeosciences, 10, 5125–5138, doi:10.5194/bg-10-5125-2013, 2013.

Hasenfratz, A.P., Martínez-García, A., Jaccard, S.L., Vance, D., Wälle, M., Greaves, and M., Haug, G.H.: Determination of the Mg/Mn ratio in foraminiferal coatings: an approach to correct Mg/Ca temperatures for Mn-rich contaminant phases, Earth Planet. Sci. Lett., 457, 335-347, https://doi.org/10.1016/j.epsl.2016.10.004, 2016.

Helmke, J.P. and Bauch, H.A.: Glacial–interglacial carbonate preservation records in the Nordic Seas, Global Planet. Change, 33, 15–28, https://doi.org/10.1016/S0921-8181(02)00058-9, 2002.

Henrich, R.: Dynamics of Atlantic water advection to the Norwegian-Greenland Sea - a time-slice record of carbonate distribution in the last 300 ky, Mar. Geol., 145, 95–131, https://doi.org/10.1016/S0025-3227(97)00103-5, 1998.

Henrich, R., Heinz Baumann, K-H., Huber, R., Meggers, H.: Carbonate preservation records of the past 3 Myr in the Norwegian–Greenland Sea and the northern North Atlantic: implications for the history of NADW production, Mar. Geol., 184, 17–39, https://doi.org/10.1016/S0025-3227(01)00279-1, 2002.

Hodell, D., Minth, E., Curtis, J., Mccave, I., Hall, I., Channell, J., and Xuan, C.: Surface and deep-water hydrography on Gardar Drift (Iceland Basin) during the Last Interglacial period, Earth Planet. Sci. Lett., 288, 10–19, https://doi.org/10.1016/j.epsl.2009.08.040, 2009.

Koho, K. A., de Nooijer, L. J., Fontanier, C., Takashi, T., Oguri, K., Kitazato, H., and Reichart, G. J.: Benthic foraminiferal Mn/Ca ratios reflect microhabitat preferences, Biogeosciences 14, 3067– 3082, https://doi.org/10.5194/bg-14-3067, 2017.

Lisiecki, L. E. and Stern, J. V.: Regional and global benthic $\delta^{18}O$ stacks for the last glacial cycle, Paleoceanography, 31, 1368–1394, https://doi.org/10.1002/2016PA003002, 2016.

Muglia, J. and Schmittner, A.: Carbon isotope constraints on glacial Atlantic meridional overturning: Strength vs depth, Quaternary Sci. Rev., 257, 106844, https://doi.org/10.1016/j.quascirev.2021.106844, 2021.

Öğretmen, N., Schiebel, R., Jochum, K. P., Galer, S., Leitner, J., Khanolkar, S., Yücel, M., Stoll, B., Weis, U., and Haug, G. H.: High precision femtosecond laser ablation ICP-MS measurement of benthic foraminiferal Mn-incorporation for paleoenvironmental reconstruction: A case study from the Plio-Pleistocene Caribbean Sea: Geochem. Geophys. Geosys., 23, https://doi.org/10.1029/2021gc010268, 2022.

PAGES – Past Interglacials Working Group: Interglacials of the last 800,000 years, Rev. Geophys., 54, 162–219, https://doi.org/10.1002/2015rg000482, 2016.

Pöppelmeier, F., Jeltsch-Thömmes, A., Lippold, J., Joos, F., and Stocker, T. F.: Multi-proxy constraints on Atlantic circulation dynamics since the last ice age, Nat. Geosci., 16, 349–356, https://doi.org/10.1038/s41561-023-01140-3, 2023.

Raitzsch, M., Hathorne, E. C., Kuhnert, H., Groeneveld, J., and Bickert, T.: Modern and late Pleistocene B/Ca ratios of the benthic foraminifer *Planulina wuellerstorfi* determined with laser ablation ICP-MS, Geology, 39, 1039–1042, https://doi.org/10.1130/G32009.1, 2011.

Rasmussen, T. L., Thomsen, E., Labeyrie, L., and van Weering, T. C. E.: Circulation changes in the Faeroe-Shetland Channel correlating with cold events during the last glacial period (58–10 ka), Geology, 24, 937, https://doi.org/10.1130/0091-7613(1996)024<0937:CCITFS>2.3.CO;2, 1996.

Teal, L. R., Bulling, M. T., Parker, E. R., and Solan, M.: Global patterns of bioturbation intensity and mixed depth of marine soft sediments, Aquat. Biol., 2, 207–218, https://doi.org/10.3354/ab00052, 2008.

Thibodeau, B., Bauch, H. A., and Pedersen, T. F.: Stratification-induced variations in nutrient utilization in the Polar North Atlantic during past interglacials, Earth Planet. Sc. Lett., 457, 127–135, 2017Thornalley, D. J. R., Bauch, H. A., Gebbie, G., Guo, W., Ziegler, M., Bernasconi, S. M., Barker, S., Skinner, L. C., and Yu, J.: A warm and poorly ventilated deep Arctic Mediterranean during the last glacial period, Science, 349, 706–710, https://doi.org/10.1126/science.aaa9554, 2015.

Trauth, M. H.: TURBO2: A MATLAB simulation to study the effects of bioturbation on paleoceanographic time series, Comput. Geosci., 61, 1–10, https://doi.org/10.1016/j.cageo.2013.05.003, 2013.

Yu, J., Menviel, L., Jin, Z. D., Thornalley, D. J. R., Barker, S., Marino, G., Rohling, E. J., Cai, Y., Zhang, F., Wang, X., Dai, Y., Chen, P., and Broecker, W. S.: Sequestration of carbon in the deep Atlantic during the last glaciation, Nat. Geosci, 9, 319–324, https://doi.org/10.1038/NGEO2657, 2016.

Zhuravleva, A., Bauch, H. A., and Van Nieuwenhove, N.: Last Interglacial (MIS5e) hydrographic shifts linked to meltwater discharges from the East Greenland margin, Quaternary Sci. Rev., 164, 95–109, https://doi.org/10.1016/j.quascirev.2017.03.026, 2017.

The paper from Stobbe et al., presents new foraminifera and pteropod data from the Norwegian Sea, covering most of the last glacial cycle. As far as I am aware this the most extensive high latitude study using B/Ca and I comment the authors for their efforts. Given its location in the central Norwegian Sea the age model seems more-or-less robust, though I would be cautious to over interpret the minutia, especially given the proxies used. I presume this is why most of the discussion is focused on MISs 4–5 where the data quality is best. The authors use the data, in context of other records to show that deep-water formation was likely a persistent feature of this region, even though an interval of major glaciation.

The paper is well presented and the data are of high quality, I have no doubt of the worthiness for publication. My comments here are mostly focused on the context and the presentation of data, which I hope will improve the manuscript.

We sincerely appreciate Thomas Chalk's thoughtful and constructive feedback. We believe that addressing these points has further strengthened the manuscript.

Detailed responses to each comment are provided below in red. Suggested changes to the main text are italicised. The line numbers refer to the revised manuscript with tracked changes. New references are given at the end of this author response letter.

Throughout the introduction I found the referencing to be a bit limited, this is especially stark when compared to the discussion (which I think is very good). It would be nice if the all papers that the data are compared to in the discussion (e.g. all the B/Ca records from table 1) were at least mentioned in the introduction, as that is really what sets up the discussion from the current state of the art.

To better establish the scientific background, we now introduce all relevant studies focussed on bottom water $[CO_3^{2-}]$ reconstructions in the Atlantic Ocean in the introduction of the revised manuscript. We hope that these additions provide a more comprehensive foundation for the analysis presented in the discussion (section 5).

I have an issue with the way the B/Ca method is presented. Especially given that many comparisons are made between datasets. The authors use JCt to normalize the B/Ca, with a value of 218 μmol/mol, but other publications have this as low as 190 μmol/mol (Hathorne et al 2013) which would make a ~10% difference. I realize the majority of the data will have used the sensitivity of *C. wuellerstorfi* to CO32- and calibrated the coretops, meaning the absolute B/Ca matters little, but is that true in every case? And it would be good regardless to make sure that the B/Ca data in its raw form is comparable in an honest way for future studies.

To make a fair comparison between different studies it is important that secondary external reference materials which are used for the normalisation stem from identical source/publication, or at the bare minimum report the source/publication, so that re-calculations are possible. It was an oversight from our side that we did not use the published element ratios by Hathorne et al. (2013) which are more widely used in the community. The value we originally used for B/Ca (217.9 μmol/mol) were based on published concentration compilation data by Jochum et al. (2019), which are based on in situ-multi element analysis data of carbonate reference materials

obtained by laser ablation. We adjusted the concentration of JCt-1 to the published values by Hathorne et al. (2013) and updated our results accordingly. This allows for more robust comparisons.

Table 1: Missing a few datasets, such as Kirby et al 2020 and Oppo et al. 2023, is there a reason these (shorter?) records are omitted from the compilation? They are both from useful areas to fill gaps.

Thank you for making us aware of these publications. We have expanded our compilation by including Lacerra et al. (2019), Kirby et al. (2020) and Oppo et al. (2023). The data are now included in Figs. 6 and 9 (original Fig. 8), while full metadata for these publications are given in Table 1.

**When comparing the datasets, I think in general the authors rely too much on their age model and maybe more importantly, those of the other studies, as well as the precision of the proxy to make detail comparisons. Alongside the analytical uncertainty. The B/Ca-CO32- proxy (Yu and Elderfield 2007) has a calibration uncertainty of ~10µmol/kg (2σ; Yu and Elderfield 2007, Yu et al., 2008; ← added by preprint authors), so a more statistical approach is warranted between some of the comparisons between sites, especially as in the literature the delta carbonate has been transformed into carbonate ion in more than one way.**

We believe that the comparison of different datasets in our study is useful, and results in important observations that are likely not compromised by the valid uncertainties mentioned by the reviewer. We have now taken care to depict a comparison of multiple datasets more reliably. For instance in section 4.3 "During MIS 5*b, 5a* and 4, bottom water [$CO_3^{2-}$] in core PS1243 are *generally higher (often by more than ~20 µmol/kg)* than reconstructed bottom water [$CO_3^{2-}$] in the Atlantic Ocean deeper than *3* km water depth (Table 1), except for Site U1313 (*~3.4 water depth*) whose [$CO_3^{2-}$] record *partly* matches *that of* PS1243 during *those times* (Fig. 6; Chalk et al., 2019). (line 611) [...] *Considering uncertainties of ~10 µmol/kg (1σ) of bottom water [$CO_3^{2-}$] reconstructions, most North Atlantic records including PS1243 seem to broadly converge during MIS 5e independent from the water depth of the core site, although slight offsets between individual records may occur (Fig. 6).(line 617)*" And "*In contrast, most Atlantic sediment cores with [$CO_3^{2-}$] estimates for the water column below 3 km water depth (Table 1) record a decrease in bottom water [$CO_3^{2-}$] from MIS 5a to MIS 4 (Fig. 6), which is consistent with Yu et al. (2016).(line 621)*" Something that we see the need to highlight are the opposite trends, namely, increasing bottom water [$CO_3^{2-}$] in shallow Atlantic cores and decreasing bottom water [$CO_3^{2-}$] in deeper cores during MIS 4, which is consistent with Yu et al. (2016). We also argue that this divergence started likely at the end of MIS 5. This is now emphasised in the revised manuscript. Accordingly, all claims of "matches" between datasets in the discussion were rephrased to "trends are similar". We hope that these clarifications suffice to address the valid comment of the reviewer.

We are somewhat hesitant to include additional illustration of these comparisons (for instance via histograms of all data in the various datasets, e.g., for the time periods MIS 4, MIS 5a-c and MIS 5e-d, for both shallow (<3 km) and deep core sites (>3 km)), as our study has already quite a high number of figures (now nine) and the existing figures already include quite a lot of data and detail. However, when the reviewer sees a strong need for such an illustration, we are very happy to include it, to support our now more nuanced assertions in the text.

Even if you age model is good enough (I am not an expert in this area), are the others good enough to draw the comparisons made on millennial timescales? I see another comment to this effect which covers my sentiments on this well.

Please see our detailed response on the interpretation of millennial timescales in our record to the major comment of Reviewer 1. We significantly amend the manuscript in lieu of these valid comments. Briefly, we have (i) substantially reduced the emphasis on millennial-scale $[CO_3^{2-}]$ variability of our study core throughout the manuscript (e.g., by significantly shortening the respective paragraph in the introduction). (ii) We remove respective statements from the abstract- and conclusions-sections of the manuscript. (iii) We incorporate reflections on millennial-scale variability in section 5.2 ("Marine Isotope Stage 5e to 5a") rather than in a separate section (previously 5.5 "Millennial-scale $[CO_3^{2-}]$ variability during Marine Isotope Stage 5"). (iv) We add limitations of our interpretations on millennial time scales in the revised manuscript, e.g., in Results section 4.2. and 4.3. and in Discussion section 5.2. And (v) we employ sediment-mixing models to quantify the effect of bioturbation on our reconstructions.

Please also note that we have removed cores EW9209-2JPC and RC16-59 from our consideration of millennial timescales, as indeed the data resolution of these cores is too low for meaningful interpretations.

Fig4: The presence of pteropods box here should be moved off axes to show the 'near zero' values in MIS4.

We adjust Fig. 4 accordingly.

I like the combination of the proxy datasets, and if the aragonite CCD is responsible for the presence and absence which is not shown in the carbonate ion records (e.g. Sulpis et al 2022) that would be very interesting. I question whether it is feasible with an epifaunal benthic species though, and do the authors have any thoughts on what you might expect the overall % aragonite to be pre-dissolution over this interval? A back of the envelope calculation for how it is changing the carbonate system would be really nice. Also, regarding the interpretation of this, is it a local effect or are you implying that pteropod dissolution is a key mechanism for increasing the carbonate ion of all the deep water emanating from this basin? As such overriding the potential of CO2 invasion and biological processes? Or should we not interpret the high CO32- values in MIS4 here outside the local area?

Assessing the impact of pteropod fluxes and -dissolution in the Nordic Seas and in the North Atlantic on the oceanic carbonate system is an important point that we agree should be assessed further. While we state that regionally pteropod dissolution might have an impact on bottom water $[CO_3^{2-}]$ at our site, and use the preservation of pteropods to highlight that the aragonite saturation depth must have deepened (supporting our *C. wuellerstorfi* B/Ca-based arguments), we feel that with the data at hand, it is impossible to estimate the impact of aragonite dissolution on bottom water $[CO_3^{2-}]$ at our site. For this endeavour, one would need to establish to what extent the observed pteropod abundances reflects changes in the pteropod downward flux, surface ocean productivity and/or changes in dissolution in bottom waters versus the water column (supralysoclinal versus below the lysocline) and how much of the downward pteropod rain has dissolved in the bottom water/porewaters at the site. These parameters are impossible to estimate from pteropod abundances alone. Given these uncertainties, we believe that any back-calculation of these effects on bottom water $[CO_3^{2-}]$ at our site would be highly speculative and we accordingly refrain from performing these calculations.

However, we cannot entirely rule out the possibility that shell dissolution elevated bottom waters at our site (and therefore mention it in the manuscript). This is something that requires investigation for the Nordic Seas and in fact the Atlantic Ocean as pteropods are very abundant pelagic organisms throughout the Atlantic Ocean (Bednaršek et al., 2012) – something that we now highlight in section 5.2. ("*However, assessing the influence of potential changes in pteropod fluxes and dissolution rates in the water column (e.g., supralysoclinal) and in bottom waters (e.g., below the lyscocline) on [CO$_3^{2-}$] in the Nordic Seas and how this might have affected the Atlantic Ocean via dense overflows requires further investigation.*")(line 896). As such, we argue that deepening of the aragonite compensation depth (ACD) and the downward transport of high-[CO$_3^{2-}$] surface waters, rather than local dissolution effects, has mostly influenced our bottom water [CO$_3^{2-}$] record, which supports studies postulating a source of well-ventilated waters emerging from the Nordic Seas during past glacials (e.g., Yu et al., 2008; Keigwin and Swift, 2017).

**Minor points**

Please be consistent with the use of hyphens (-) and 'n-dashes' (–) throughout, at the moment they are mixed. 'N-dashes' should be used for all ranges.

Corrected.

Line 9: Simplify to say 'higher CO32- values in MIS4 and 5 than the Holocene'.

Corrected.

28: careful when talking about a 'transition' with two non-adjacent stages. I would prefer 'between' or similar when talking about an extended time period.

Corrected.

43: denser water masses, it's all relative.

Corrected.

55: Crocker et al 2016 could be added as a reference here (see point above).

We have rewritten the introduction in line with our reply to the first comment of the reviewer and included the reference of Crocker et al. (2016) in the introduction too (line 94).

76: please give values as well as the CO2 decline.

We provide both absolute values for atmospheric CO$_2$ change and the magnitude of decline.

116: AF, defined by s=35

Corrected.

165: I might be out of date here, but I thought the value was globally closer to 0.48 (Marchitto et al 2014) is there a reason that locally 0.64 works better here?

We are aware of studies re-evaluating the disequilibrium factor for *C. wuellerstorfi* $\delta^{18}O$ and seawater $\delta^{18}O$ (which includes using *Uvigerina* $\delta^{18}O$ as reference). Some studies find indeed a difference between the two of 0.47 ‰ (Marchitto et al., 2014), others support the original value of 0.64 ‰ (Jöhnck et al., 2021). This is a complex issue, as $\delta^{18}O$ disequilibria likely vary spatially and temporally, and its estimation varies between studies and requires a number of assumptions. However, the choice of the correction factor has no impact on any of our results and discussions. We therefore chose to apply the originally reported value of Shackleton (1974) as this is consistent with factors used for correcting *C. wuellerstorfi* $\delta^{18}O$ in our study core PS1243 in earlier publications (Bauch et al., 2001; Bauch and Erlenkeuser, 2003; Bauch et al., 2012).

196: I always worry about this use of multiple samples to represent a stable late Holocene situation, is it valid? Are the uncertainties propagated through the rest of the record? 1.4µmol/mol seems very low for a measurement error combined with a n=3 grouping.

The 1.4 µmol/mol (revised to 1.2 µmol/mol after the revised normalisation, see our reply to the comment on data normalisations) represents the standard deviation specifically for the n=3 grouping of the three uppermost *C. wuellerstorfi* B/Ca measurements. We believe that grouping these three measurements is a reliable approach to determine a core-top value rather than relying on a single datapoint that may be an outlier. Along with our measurement error of 4.3 µmol/mol, we calculate an 1σ-uncertainty of the core-top B/Ca of 4.5 µmol/mol. We additionally have now added a more detailed error propagation of the total uncertainty for the $[CO_3^{2-}]_{down-core}$ record to the manuscript: "*The total uncertainty of 9.3 µmol/kg (1σ) for our B/Ca-derived $[CO_3^{2-}]_{down-core}$ values results from 1σ-uncertainties for $[CO_3^{2-}]_{pre-industrial}$ of 5.6 µmol/kg, for $B/Ca_{down-core}$ of 4.3 µmol/mol, for $B/Ca_{core-top}$ of 4.5 µmol/mol, as well as the calibration error of 5 µmol/kg (1σ; Yu and Elderfield, 2007; Yu et al., 2008).[...](section 3.3; line 304)*" Further, we added uncertainty bars (1σ) in revised Figs. 3, 4, 6, 7 and the newly added Fig. 8 to be transparent about the uncertainties of our proxy reconstructions (please refer to the appendix).

200: the use of 10±5 seems a bit arbitrary here, is it just somewhere in between the other estimates? Would you explain better the process of finding this number?

We more thoroughly state the process of determining the correction factor for anthropogenic $CO_2$ evasion and added a correction: "Because Olsen et al. (2010) and Vázquez-Rodríguez et al. (2009) estimated an anthropogenic dissolved inorganic carbon (DIC) addition of ~8±3 µmol/kg (2σ) and ~12±10 µmol/kg (2σ) for the deep Nordic Seas, respectively, *we calculate a mean* anthropogenic DIC contribution of 10±6 µmol/mol (2σ) for bottom waters at our study site *from both of these estimates.*" (section 3.3; line 287). We further specify: "*This implies a $[CO_3^{2-}]$ decrease of ~5±3 µmol/kg (2σ) due to the evasion of anthropogenic $CO_2$ into the deep Norwegian Sea [...]*" (section 3.3; line 294), and hope that the reviewer finds these specifications informative and clear.

284: See above, but stating that PS1243 is 'consistently' higher seems like an overstatement. Slightly higher?

Corrected. We now state: "During *late* MIS 5 and MIS 4, *C. wuellerstorfi* B/Ca-derived $[CO_3^{2-}]$ at site PS1243 are *generally* higher than during the Holocene ([…]) and $[CO_3^{2-}]_{pre-industrial}$ at the core site ([…]). During MIS 5e ([…]) *and 5d ([...]), reconstructed bottom water $[CO_3^{2-}]$ estimates in the deep Norwegian Sea are only slightly higher than the Holocene mean, yet these differences are statistically significant within 95% confidence level based on a two-sided*

*student t-test (p<0.01; Fig. 4).*" (section 4.2; line 511). We believe that this provides a more nuanced representation of our data.

291: contextualise "low CO2" MIS 5b (and d) is still an interglacial interval.

We rephrase our statement: "[…] both *C. wuellerstorfi* $\delta^{13}C$ and bottom water $[CO_3^{2-}]$ show elevated values *despite a lowering of $CO_{2,atm}$ (Figs. 4, 5)*" (section 4.2; line 519).

304: here you say the variations are similar to MD95-2039 but above you say significantly offset. Explain more precisely the situation.

We clarify these ambiguities and state "On average, bottom water $[CO_3^{2-}]$ variations *obtained from core PS1243 are offset by 22±9 µmol/kg from bottom water $[CO_3^{2-}]$ reconstructed in* Iberian margin core MD95-2039 during MIS 5e and 5d (Fig. 4; *Yu et al., 2023).*" (section 4.2; line 571).

Fig 5: The colour scale in the middle makes it look related to the y-axis. Would work better as a traditional legend.

We use a separate legend for the colour scale in revised Fig. 5 that is detached from the data panels.

329: I'm not sure this point is borne out by the data, at least as presented here. Perhaps some statistics or an additional figure (histogram of values?) could help? U1308 and RC 16-59 data also look like they are in approximately the same area and are both >2km depth.

We adjusted our statement to emphasise differences in bottom water $[CO_3^{2-}]$ records during MIS 5b, 5a and 4 that show a general divergence, and during MIS 5e and 5d that generally indicate a convergence of $[CO_3^{2-}]$ records. We now state (section 4.3; line 611): „During MIS *5b, 5a* and 4, bottom water $[CO_3^{2-}]$ in core PS1243 are *generally* higher *(often by more than ~20 µmol/kg)* than reconstructed bottom water $[CO_3^{2-}]$ in the Atlantic Ocean deeper than 3 km water depth (Table 1), except for Site U1313 (3.4 km water depth) whose […]" and "*Considering uncertainties of ~10 µmol/kg (1σ) of bottom water $[CO_3^{2-}]$ reconstructions, most North Atlantic $[CO_3^{2-}]$ records including PS1243 seem to broadly converge during MIS 5e independent from the water depth of the core site, although slight offsets between individual records may occur (Fig. 6).*"(line 617). We hope that this more carefully reflects the nuances of all datasets in this case.

336: U1313 is in the western Atlantic basin, not the east.

We correct this and now state instead "*central North Atlantic (U1308 and U1313)*" (section 4.3; line 623).

Figure 6: This figure is a bit strangely organized, I'm not sure why the MD95 and PS1243 records both appear twice, and in the case of the latter with different symbols. The mixture of carbonate ion and delta carbonate ion (an estimation of CO32-sat could be made?) as well as the separation of the panels without labelling why they are divided so makes it difficult to read.

Also in response to reviewer 1, we improved Fig. 6: (i) We rearrange time series according to their geographic location, separate them on different axes (e.g., North Atlantic, tropical Atlantic and South Atlantic) and clearly labelled this separation, (ii) we convert $\Delta[CO_3^{2-}]$ to $[CO_3^{2-}]$ for

the GeoB cores of Raitzsch et al. (2011) following Yu et al. (2016), (iii) we clarify in the figure caption that the comparison of the records from PS1243 and MD95-2039 is shown on a separate axis "to aid a comparison on (multi-)millennial-scale timescales", (iv) lastly, we have removed full core names from the map and replaced them with core numbers (referencing to full metadata information in Table 1 for a better overview). Symbols in the map are identical to those of the timeseries.

362: I'm not sure that the arguments on this are so one-sided? See Gallaasen et al. 2014 and others, the geochemical studies cited here are consistent with multiple potential export strengths, as they are looking at stratification and water sources.

We adjust our wording to highlight that we specifically refer to the mean state of the Holocene and MIS 5e, when mentioning $CO_{2,atm}$ levels (Bereiter et al., 2015) and the NADW export strength (Böhm et al., 2015). To avoid confusion with other geochemical parameters, we only cite Böhm et al. (2015) when referring to NADW export strength. Additionally, also in light of the important comment of Reviewer 1, we specify in the same section 5.1 that "our new bottom water $[CO_3^{2-}]$ data suggest that *these additional influences on Nordic Seas surface ocean buoyancy* did not hamper deep convection in the Norwegian Sea *beyond centuries or few (sub-)millennia"* ( line 782)., citing the study of Galaasen et al. (2014) further up. This should provide some nuance to our statement and leaves open the possibility of high-frequency variations in MIS 5e.

379: growth appears twice.

Corrected.

403: remove transition here.

We removed the word "transition" in the heading of section 5.2. and 5.3.

410: see new temperature estimates in Morley et al 2024, which shows even larger T changes.

We prefer to not cite the study of Morley et al. (2024), as this study focusses on the LGM and offers no constraints on the MIS 5e/d boundary. Also, for our considerations in section 5.2 it is not strictly important what the exact magnitude of cooling was from MIS 5e to MIS 5d.

487: you have und in place of and.

We corrected all instances of the German word "und" throughout the manuscript.

Fig 8: This figure is a bit bizarre, I think it needs to be bigger and have the map placed above or below the data for clarity. The colour palette could also be used to aid interpretation instead of the (random?) selection currently used. The yellow is difficult to see, and it took me a while to notice the x-axes are not equally scaled. If recalibrating data (e.g. 980) why not use the higher resolution data from Crocker et al. 2016 also? And maybe picked a preferred calibration.

Also in response to a comment of reviewer 1, we improve new Fig. 9 (original Fig. 8) as follows: (i) we illustrate core locations in two separate maps depending on the water depth of the core site (above and below 3 km) as this also separates general trends in seawater $[CO_3^{2-}]$ changes. (ii) We replace the rather long core labels in the map with numbers (that cross-reference to the metadata of the core sites in existing Table 1). (iii) We increase x-axes lengths and place the

deep (>3 km) and shallow cores (<3 km) on different y-axes to enhance clarity. (iv) We equally scaled the x-axes. (v) We use a colour coding from north (warm colours) to south (cold colours) for time series and symbols. We hope that these changes sufficiently enhance the clarity of the figure.

Nonetheless, we also want to emphasise that it is not the detail of a single time series or the relationship of every individual time series towards each other that is important. Rather, the main message are the opposing trends in $[CO_3^{2-}]$ changes in cores identified shallower or deeper than 3 km water depth, which we hope becomes sufficiently clear from revised Fig. 9 (original Fig. 8).

We now also show data sets of Chalk et al. (2019) (solid line in new Fig. 9; appendix) and Crocker et al. (2016) (stippled line in new Fig. 9; appendix) separately. We converted the *C. wuellerstorfi* B/Ca record from Crocker et al. (2016) into a $[CO_3^{2-}]_{\text{down-core}}$ record using the same conversion applied in our manuscript ($[CO_3^{2-}]_{\text{down-core}} = [CO_3^{2-}]_{\text{pre-industrial}} + (\text{B/Ca}_{\text{down-core}} - \text{B/Ca}_{\text{core-top}}) / 1.14$; section 3.3) using the core-top $[CO_3^{2-}]$ value (119 µmol/kg) reported by Chalk et al. (2019) and the uppermost B/Ca value of Crocker et al. (2016) to represent the core-top value.

562: more or deeper records required, as U1313 is already W. Atlantic.

We specify our statement accordingly: "*However, unravelling NSSW $[CO_3^{2-}]$ variability* […] *requires high-resolution bottom water $[CO_3^{2-}]$ records from high-sedimentation sites in the Nordic Seas as well as from deep Atlantic sites (e.g., from the deep western basin of the Atlantic Ocean).*" (now section 5.2; line 929).

References:

Bauch, H. A. and Erlenkeuser, H.: Interpreting Glacial-Interglacial Changes in Ice Volume and Climate From Subarctic Deep Water Foraminiferal δ18O, in: Earth's Climate and Orbital Eccentricity: The Marine Isotope Stage 11 Question, Geoph. Monog. Series 137, edited by: Droxler, L. H., Poore, A. W., and Burckle, R. Z., American Geophysical Union, Washington, D.C., 87–102, 2003.

Bauch, H. A., Erlenkeuser, H., Spielhagen, R. F., Struck, U., Matthiessen, J., Thiede, J., and Heinemeier, J.: A multiproxy reconstruction of the evolution of deep and surface waters in the subarctic Nordic seas over the last 30,000 yr, Quaternary Sci. Rev., 20, 659–678, https://doi.org/10.1016/S0277-3791(00)00098-6, 2001.

Bauch, H. A., Kandiano, E. S., and Helmke, J. P.: Contrasting ocean changes between the subpolar and polar North Atlantic during the past 135 ka, Geophys. Res. Lett., 39, https://doi.org/10.1029/2012GL051800, 2012.

Bednaršek, N., Mozina, J., Vogt, M., O'Brien, C. and Tarling, G.A.: The global distribution of pteropods and their contribution to carbonate and carbon biomass in the modern ocean, Earth Syst. Sci. Data, 4, 167–186, https://doi.org/10.5194/essd-4-167-2012.

Bereiter, B., Eggleston, S., Schmitt, J., Nehrbass-Ahles, C., Stocker, T. F., Fischer, H., Kipfstuhl, S., and Chappellaz, J.: Revision of the EPICA Dome C $CO_2$ record from 800 to 600 kyr before present, Geophys. Res. Lett., 42, 542–549, https://doi.org/10.1002/2014GL061957, 2015.

Böhm, E., Lippold, J., Gutjahr, M., Frank, M., Blaser, P., Antz, B., Fohlmeister, J., Frank, N., Andersen, M. B., and Deininger, M.: Strong and deep Atlantic meridional overturning circulation during the last glacial cycle, Nature, 517, 73–76, https://doi.org/10.1038/nature14059, 2015.

Chalk, T. B., Foster, G. L., and Wilson, P. A.: Dynamic storage of glacial $CO_2$ in the Atlantic Ocean revealed by boron [$CO_3^{2-}$] and pH records, Earth Planet. Sc. Lett., 510, 1–11, https://doi.org/10.1016/j.epsl.2018.12.022, 2019.

Crocker, A. J., Chalk, T. B., Bailey, I., Spencer, M. R., Gutjahr, M., Foster, G. L., and Wilson, P. A.: Geochemical response of the mid-depth Northeast Atlantic Ocean to freshwater input during Heinrich events 1 to 4, Quaternary Sci. Rev., 151, 236–254, https://doi.org/10.1016/j.quascirev.2016.08.035, 2016.

Galaasen, E. V., Ninnemann, U. S., Irvalı, N., Kleiven, H. K. F., Rosenthal, Y., Kissel, C., and Hodell, D. A.: Rapid reductions in North Atlantic Deep Water during the peak of the last interglacial period, Science, 343, 1129–1132, 2014.

Hathorne, E., Gagnon, A., Felis, T., Adkins, J., Asami, R., Boer, W., Caillon, N., Case, D., Cobb, K. M., Douville, E., deMenocal, P., Eisenhauer, A., Garbe-Schönberg, D., Geibert, W., Goldstein, S., Hughen, K., Inoue, M., Kawahata, H., Kölling, M., Cornec, F. L., Linsley, B. K., McGregor, H. V., Montagna,P., Nurhati, I. S., Quinn, T. M., Raddatz, J., Rebaubier, H., Robinso, L., Sadekov, A., Sherrell, R., Sinclair, D., Tudhope, A. W., Wei, G., Wong, H., Wu, H. C., and You, C.-F.: Interlaboratory study for coral Sr/Ca and other element/Ca ratio measurements, Geochem. Geophy. Geosy., 14, 3730–3750, https://doi.org/10.1002/ggge.20230, 2013.

Jochum, K. P., Garbe-Schönberg, D., Veter, M., Stoll, B., Weis, U., Weber, M., Lugli, F., Jentzen, A., Schiebel, R., Wassenburg, J. A., Jacob, D. E., and Haug, G. H.: Nano-Powdered Calcium Carbonate Reference Materials: Significant Progress for Microanalysis?, Geostand. Geoanal. Res., 43, 595–609, https://doi.org/10.1111/ggr.12292, 2019.

Jöhnck, J., Holbourn, A. E., Kuhnt, W., and Andersen, N.: Oxygen isotope offsets in deep-water benthic foraminifera, J. Foramin. Res., 51, 225–244, https://doi.org/10.2113/gsjfr.51.3.225, 2021.

Keigwin, L. D. and Swift, S. A.: Carbon isotope evidence for a northern source of deep water in the glacial western North Atlantic, P. Natl. ACAD. Sci. USA, 114, 2831–2835, https://doi.org/10.1073/pnas.1614693114, 2017.

Kirby, N., Bailey, I., Lang, D.C., Brombacher, A., Chalk, T.B., Parker, R.L., Crocker, A.J., Taylor, V.E., Milton, J.A., Foster, G.L., Raymo, M. E., Kroon, D., Bell, D. B., and Wilson, P. A.: On climate and abyssal circulation in the Atlantic Ocean during late Pliocene marine isotope stage M2, ~ 3.3 million years ago, Quaternary Sci. Rev., 250, 106644, https://doi.org/10.1016/j.quascirev.2020.106644, 2020.

Lacerra, M., Lund, D., Gebbie, G., Oppo, D. W., Yu, J., Schmittner, A., and Umling, N. E.: Less remineralized carbon in the intermediate-depth South Atlantic during Heinrich Stadial 1, Paleoceanography, 34, 1218-1233, https://doi.org/10.1029/2018PA003537, 2019.

Marchitto, T., Curry, W., Lynch-Stieglitz, J., Bryan, S., Cobb, K., and Lund, D.: Improved oxygen isotope temperature calibrations for cosmopolitan benthic foraminifera, Geochim. Cosmochim. Acta, 130, 1–11, https://doi.org/10.1016/j.gca.2013.12.034, 2014.

Morley, A., de la Vega, E. and Raitzsch, M.: A solution for constraining past marine Polar Amplification. Nature Commun., 15, 9002, https://doi.org/10.1038/s41467-024-53424-w, 2024.

Olsen, A., Omar, A. M., Jeansson, E., Anderson, L. G., and Bellerby, R. G. J.: Nordic seas transit time distributions and anthropogenic $CO_2$, J. Geophys. Res., 115, https://doi.org/10.1029/2009JC005488, 2010.

Oppo, D. W., Lu, W., Huang, K.-F., Umling, N. E., Guo, W., Yu, J., Curry, W. B., Marchitto, T. M., and Wang, S.: Deglacial Temperature and Carbonate Saturation State Variability in the Tropical Atlantic at Antarctic Intermediate Water Depths, Paleoceanography, 38, https://doi.org/10.1029/2023PA004674, 2023.

Raitzsch, M., Hathorne, E. C., Kuhnert, H., Groeneveld, J., and Bickert, T.: Modern and late Pleistocene B/Ca ratios of the benthic foraminifer *Planulina wuellerstorfi* determined with laser ablation ICP-MS, Geology, 39, 1039–1042, https://doi.org/10.1130/G32009.1, 2011.

Vázquez-Rodríguez, M., Touratier, F., Lo Monaco, C., Waugh, D. W., Padin, X. A., Bellerby, R. G. J., Goyet, C., Metzl, N., Ríos, A. F., and Pérez, F. F.: Anthropogenic carbon distributions in the Atlantic Ocean: data-based estimates from the Arctic to the Antarctic, Biogeosciences, 6, 439–451, https://doi.org/10.5194/bg-6-439-2009, 2009.

Yu, J. and Elderfield, H.: Benthic foraminiferal B/Ca ratios reflect deep water carbonate saturation state, Earth Planet. Sc. Lett., 258, 73–86, https://doi.org/10.1016/j.epsl.2007.03.025, 2007.

Yu, J., Elderfield, H., and Piotrowski, A. M.: Seawater carbonate ion-$\delta^{13}C$ systematics and application to glacial–interglacial North Atlantic Ocean circulation, Earth Planet. Sc. Lett., 271, 209–220, https://doi.org/10.1016/j.epsl.2008.04.010, 2008.

Yu, J., Menviel, L., Jin, Z. D., Thornalley, D. J. R., Barker, S., Marino, G., Rohling, E. J., Cai, Y., Zhang, F., Wang, X., Dai, Y., Chen, P., and Broecker, W. S.: Sequestration of carbon in the deep Atlantic during the last glaciation, Nat. Geosci, 9, 319–324, https://doi.org/10.1038/NGEO2657, 2016.

**Authors response to the Editors decision**

Public justification (visible to the public if the article is accepted and published):

The authors have done a thorough job responding to the reviewers' comments and are encouraged to implement all changes described.

Additionally, I have a couple comments about the age model for core PS1243. My first concern is the choice to align the planktonic d18O record for this core to a benthic d18O stack. The potential problem is that planktonic d18O at this site has negative excursions during Heinrich Stadials (HS), whereas the benthic d18O stack has positive excursions during these events. The secondary alignment of identified HS to the corresponding events in NGRIP will largely address most of the bias that may have occurred from alignment to benthic d18O. However, the alignment of MIS5e might still be affected. Could the authors address whether the early peak in planktonic d18O aligned to the start of MIS5e might actually be related to a HS during Termination 2 (analogous to the planktonic d18O peaks for the Younger Dryas and HS1)?

Although *N. pachyderma* $\delta^{18}$O has proven to adequately reflect glacial-interglacial climate variability in the North Atlantic and in the Nordic Seas (e.g., Bauch et al., 2012; Obrochta et al., 2014), we agree with the editor about potential caveats of aligning a planktonic $\delta^{18}$O record to the benthic $\delta^{18}$O stack on millennial timescales. For that reason, we have chosen a fine-tuning based on residual planktonic $\delta^{18}$O minima and corresponding NGRIP events (Figure 2). The tiepoint at the MIS 6-5 boundary and our labelling of MIS 5e in the figures might be indeed misleading regarding the relationship between *N. pachyderma* $\delta^{18}$O minima and HS11. We now define the onset of MIS 5e in the Nordic Seas after Bauch et al. (2012) – also referred to as a regional MIS 5e *sensu stricto* - marked by the end of ice-rafted debris input to the study site (i.e., at 240 cm). We have adjusted the labelling of MIS 5e in all figures and state our definition of the MIS 5e onset in the figure captions ("The onset of MIS5e was defined in the study core after Bauch et al. (2012) as the end of ice-rafted detritus input following MIS 6"). This implies that the *N. pachyderma* $\delta^{18}$O minimum falls indeed into Termination II (i.e., meltwater overprint), as the editor surmised. The main text of the manuscript was revised to reflect the slightly shorter MIS 5e after our newly chosen definition consistent with Bauch et al. (2012). The exact onset of MIS5e (original versus revised version) does not change the conclusions of our study in any way.

My second concern is that names for HS before HS6 are not standardized in the literature. If this manuscript is following the naming convention of a previous study, that should be explained/cited. Otherwise, the authors should consider following the convention used for their alignment target, Lisiecki and Stern (2016). The event labeled HS10 in this manuscript was called HS9 in Lisiecki and Stern (2016), which may cause confusion for some readers. One option would be to relabel the event currently called HS9 to HS8a and H10 to H9. Alternatively, the authors should explain the naming discrepancy in the text and/or the caption to Figure 2.

We appreciate the editor for bringing to our attention differences in the naming convention of HS before HS6. In our manuscript, we followed the naming convention of Yu et al. (2023) for the HS, as the analyses in core MD95-2039 (Yu et al., 2023) form a crucial basis for comparison with our study

core. To ensure clarity, we have added information on our HS naming convention to the revised manuscript accordingly and have highlighted potential differences in other studies, where necessary, i.e., in the figure captions of Figs. 2, 3, 4, 6 and 8: "HS numbering follows Yu et al. (2023), which may differ from the naming convention in other studies, especially for HS older than HS6."

References:

Bauch, H. A., Kandiano, E. S., and Helmke, J. P.: Contrasting ocean changes between the subpolar and polar North Atlantic during the past 135 ka, Geophys. Res. Lett., 39, https://doi.org/10.1029/2012GL051800, 2012.

Obrochta, S.P., Crowley, T.J., Channell, J.E.T., Hodell, D.A., Baker, P.A., Seki, A., Yokoyama, Y.: Climate variability and ice-sheet dynamics during the last three glaciations. Earth Planet. Sci. Lett. 406, 198–212. https://doi.org/10.1016/j.epsl.2014.09.004, 2014.

Yu, J., Anderson, R. F., Jin, Z. D., Ji, X., Thornalley, D. J. R., Wu, L., Thouveny, N., Cai, Y., Tan, L., Zhang, F., Menviel, L., Tian, J., Xie, X., Rohling, E. J., and McManus, J. F.: Millennial atmospheric CO2 changes linked to ocean ventilation modes over past 150,000 years, Nat. Geosci, 16, 1166–1173, https://doi.org/10.1038/s41561-023-01297-x, 2023.

---

## Author Response (AR2)

**Final authors response**

**The authors have done a thorough job responding to reviewer comments. After a few technical corrections, this manuscript is ready for publication.**

We sincerely thank Lorraine Lisiecki and Thomas Chalk for accepting our revised manuscript for publication and for their valuable suggestions. We are confident that the final revisions have provided the manuscript with the necessary final polish.

Detailed responses to each comment are provided below in red. Suggested changes to the main text are indicated in *italics*.

**Technical corrections:**

**There are many extraneous-seeming hyphens that should be removed (e.g., lines 49, 103, 109, 404, 409, 631**

We have carefully corrected the text and removed all extraneous hyphens, including those noted in lines 49, 103, 109, 404, 409, and 631.

**Instead of including the sentences "HS numbering follows Yu et al. (2023), which may differ from the naming convention in other studies, especially for HS older than HS6. The onset of MIS 5e (*sensu stricto*) was defined in the study core after Bauch et al. (2012) as the end of ice-rafted detritus input following MIS 6."repeatedly in figure captions, please include these explanations in section 3.5 "Age Model" plus the sentence about HS numbering in only the Figure 2 caption.**

We have removed the sentence from all figure captions except for Figure 2 and have included the following text at line 257 in Section 3.5: *We adopt the HS numbering from Yu et al. (2023), noting that it may differ from the naming conventions applied in other studies, particularly for HS older than HS6.*

**Add the demarcation for the start of MIS 5e (*sensu stricto*) to Table 2, so it is clear which depth was identified as the end of IRD and what age it has been assigned on your age model. The tuning target could be listed as "interpolated."**

We have included the onset of MIS 5e (*sensu stricto*) in Table 2, indicating the corresponding depth and assigned age from our age model. The tuning target is listed as "interpolated" as suggested.

**Consider adding brief y-axis labels, such as ("Mn/Ca" and "Al/Ca") to the subplots in figure 3 (the cross plots), if you can find a way to without increasing the clutter.**

We have added y-axis labels ("Mn/Ca ($\mu$mol mol$^{-1}$)" and "Al/Ca ($\mu$mol mol$^{-1}$)") to the subplots in Figure 3 as suggested, ensuring that the figure remains clear and uncluttered.

**Line 420+: rephrase to "multiple lines of evidence for", and either expand or rephrase what is meant precisely by "late thermal maximum" (e.g. why would you consider it late?).**

We adjust our statement accordingly: There *are* also multiple *lines of* evidence for prolonged meltwater supply from remnant MIS 6 ice sheets surrounding the Nordic Seas which extended far into MIS 5e (Bauch et al., 2011, 2012) that may have been linked with centennial- to sub-millennial-scale disruptions in NADW formation (e.g. Galaasen et al., 2014). We have decided to remove the sentence "[…] causing a late thermal maximum in the upper ocean (Bauch et al., 2011; Capron et al., 2014; Zhuravleva et al., 2017) [...]." as it is not strictly relevant to the discussion at this line, hence we also removed the statement on "late thermal maximum". For completeness though, it refers to the delayed sea surface temperature maximum in MIS 5e in the Nordic Seas compared to North Atlantic records (Bauch et al., 2011; Capron et al., 2014; Zhuravleva et al., 2017). This is mentioned in line 435 anyway.

**Line 425: The phrase "centuries or few (sub-)millennia" is overly ambiguous. Change "few" to "several" and clarify if you mean sub-millennia or millennia.**

As we have demonstrated that our record resolves millennial-scale variability, we now state: Our new bottom water [$CO_3^{2-}$] data suggest that these additional influences on Nordic Seas surface ocean buoyancy did not hamper deep convection in the Norwegian Sea beyond *several* centuries or *very few millennia.*

**Line 429: It isn't clear what "respectively" applies to in this sentence.**

We recognised that our initial statement was misleading and removed "respectively" from the text. It now reads: Indeed, modelling studies forced by various future $CO_{2,atm}$ scenarios suggest a weakening of the AMOC during the first century and a decline in overturning in the Nordic Seas until ~2050 that is followed by an AMOC recovery to the initial or an even stronger state (depending on the model; Bonan et al., 2022) and a strengthening of overturning in the Nordic Seas (Årthun et al., 2023).